# Profiling sensory neuron microenvironment after peripheral and central axon injury reveals key pathways for neural repair

**Oshri Avraham[1], Rui Feng[1], Eric Edward Ewan[1], Justin Rustenhoven[2,3], Guoyan Zhao[1,2], Valeria Cavalli[1,4,5]\***

[1]Department of Neuroscience, Washington University School of Medicine, Saint Louis, United States; [2]Department of Pathology and Immunology, Washington University School of Medicine, St Louis, United States; [3]Center for Brain Immunology and Glia (BIG), Washington University School of Medicine, St Louis, United States; [4]Center of Regenerative Medicine, Washington University School of Medicine, St. Louis, United States; [5]Hope Center for Neurological Disorders, Washington University School of Medicine, St. Louis, United States

**ABSTRACT** Sensory neurons with cell bodies in dorsal root ganglia (DRG) represent a useful model to study axon regeneration. Whereas regeneration and functional recovery occurs after peripheral nerve injury, spinal cord injury or dorsal root injury is not followed by regenerative outcomes. Regeneration of sensory axons in peripheral nerves is not entirely cell autonomous. Whether the DRG microenvironment influences the different regenerative capacities after injury to peripheral or central axons remains largely unknown. To answer this question, we performed a single-cell transcriptional profiling of mouse DRG in response to peripheral (sciatic nerve crush) and central axon injuries (dorsal root crush and spinal cord injury). Each cell type responded differently to the three types of injuries. All injuries increased the proportion of a cell type that shares features of both immune cells and glial cells. A distinct subset of satellite glial cells (SGC) appeared specifically in response to peripheral nerve injury. Activation of the PPARα signaling pathway in SGC, which promotes axon regeneration after peripheral nerve injury, failed to occur after central axon injuries. Treatment with the FDA-approved PPARα agonist fenofibrate increased axon regeneration after dorsal root injury. This study provides a map of the distinct DRG microenvironment responses to peripheral and central injuries at the single-cell level and highlights that manipulating non-neuronal cells could lead to avenues to promote functional recovery after CNS injuries or disease.

**\*For correspondence:**
cavalli@wustl.edu

**Competing interest:** The authors declare that no competing interests exist.

## Introduction

Peripheral sensory neurons activate a pro-regenerative program after nerve injury to enable axon regeneration and functional recovery. In contrast, axons fail to regenerate after central nervous system injury, leading to permanent disabilities. Sensory neurons with cell bodies in dorsal root ganglia (DRG) represent one of the most useful models to study axon regeneration. Sensory neurons send a single axon which bifurcates within the ganglion; one axon proceeds centrally along the dorsal root into the spinal cord and the other proceeds along peripheral nerves. Whereas regeneration and functional recovery can occur after peripheral nerve injury, dorsal root injury, or spinal cord injury is not followed by regenerative outcomes (*Attwell et al., 2018*; *He and Jin, 2016*; *Mahar and Cavalli, 2018*; *Tran et al., 2018*). This results in part from a failure of central injury to elicit a pro-regenerative response

in sensory neurons (*Attwell et al., 2018*; *Fagoe et al., 2014*; *Mahar and Cavalli, 2018*; *Tran et al., 2018*).

The dorsal root injury is a useful model to understand how to promote axon growth into the central nervous system (*Smith et al., 2012*). Dorsal root disruption can occur in brachial plexus injuries, leading to paralysis of the affected arm (*Smith et al., 2012*). Regeneration following dorsal root crush can occur along the growth-supportive environment of Schwann cells, but stops as the axons reach the transition between the peripheral nervous system and the central nervous system, termed the dorsal root entry zone, where a variety of inhibitory factors block further growth (*Smith et al., 2012*). However dorsal root axonal growth occurs only at half the rate of peripheral axons (*Oblinger and Lasek, 1984*; *Wujek and Lasek, 1983*). The histological difference between dorsal roots and peripheral nerve is not sufficient to alter the rate of axonal regeneration (*Wujek and Lasek, 1983*). Rather, the availability of trophic factors and other target derived influences via the peripheral axon were suggested to prevent the upregulation of pro-regenerative genes such as *Jun* (*Broude et al., 1997*) or *Gap43* (*Schreyer and Skene, 1993*). Interestingly, dorsal root injury causes up-regulation of the pro-regenerative gene *Atf3* (*Huang et al., 2006*), but only in large diameter neurons, whereas *Atf3* and *Jun* are upregulated in a majority of neurons after peripheral nerve injury (*Chandran et al., 2016*; *Renthal et al., 2020*; *Seijffers et al., 2007*; *Tsujino et al., 2000*). Spinal cord injury also leads to activation of *Atf3* in large diameter neurons, but this is not sufficient to promote regenerative growth (*Ewan et al., 2021*). Another possibility explaining the slow growth capacity of axons in the injured dorsal root is the contribution of non-neuronal cells.

Regeneration of axons in peripheral nerves is not cell autonomous. At the site of injury in the nerve, Schwann cells (*Jessen and Mirsky, 2016*) and macrophages (*Zigmond and Echevarria, 2019*) contribute to promote axon regeneration. In the DRG, macrophages are involved in eliciting a pro-regenerative response after peripheral but not central injury (*Kwon et al., 2013*; *Niemi et al., 2016*; *Niemi et al., 2013*; *Zigmond and Echevarria, 2019*), with anti-inflammatory macrophages believed to be more involved in the regenerative process than pro-inflammatory macrophages (*Zigmond and Echevarria, 2019*). We recently revealed that satellite glial cells (SGC), which completely surround sensory neuron soma, also contribute to promote axon regeneration (*Avraham et al., 2020*). PPARα signaling downstream of fatty acid synthase (FASN) in SGC promote axon regeneration in peripheral nerves, in part via regulating the expression of pro-regenerative genes in neurons, such as *Atf3* (*Avraham et al., 2020*). Whether the different regenerative capacities after peripheral or central axon injury result, at least in part from a lack or an altered response of macrophages, SGC or other non-neuronal cells in the DRG microenvironment remains largely unknown.

To answer this question, we performed a comprehensive single-cell transcriptional profiling of DRG cells after peripheral injury (sciatic nerve crush) and central injuries (dorsal root crush and spinal cord injury). Sciatic nerve crush injures approximately half the axons projecting into the peripheral nerves (*Laedermann et al., 2014*; *Rigaud et al., 2008*) and dorsal root crush injures all axons projecting into the dorsal root. Dorsal column lesion of the spinal cord damages the ascending axon branches of most large diameter neurons and leaves the descending axon branches in the spinal cord intact (*Attwell et al., 2018*; *Niu et al., 2013*; *Zheng et al., 2019*). We found that gene expression changes occurred in endothelial cells, pericytes, Schwann cells, macrophages and SGC after peripheral nerve injury, but also occurred after dorsal root crush and spinal cord injury. However, each cell type responded differently to each injury. We show that SGC fail to activate the PPARα signaling pathway in response to dorsal root crush and downregulate this pathway in response to SCI. Using the PPARα agonist fenofibrate, an FDA-approved compound used to treat dyslipidemia (*Rosenson, 2008*), axon regeneration after dorsal root crush was increased. These results indicate that the DRG microenvironment respond differently to central and peripheral axon injuries and that manipulating non-neuronal cells could lead to avenues to promote functional recovery after CNS injuries. Our study establishes a resource for understanding the functions of non-neuronal cells in the dorsal root ganglia across different types of injuries. Our data is available on a web-based resource for exploring changes in gene expression in DRG cells after central and peripheral injuries (https://mouse-drg-injury.cells.ucsc.edu/), which will aid the field to study the role of the DRG microenvironment in functional recovery after injury.

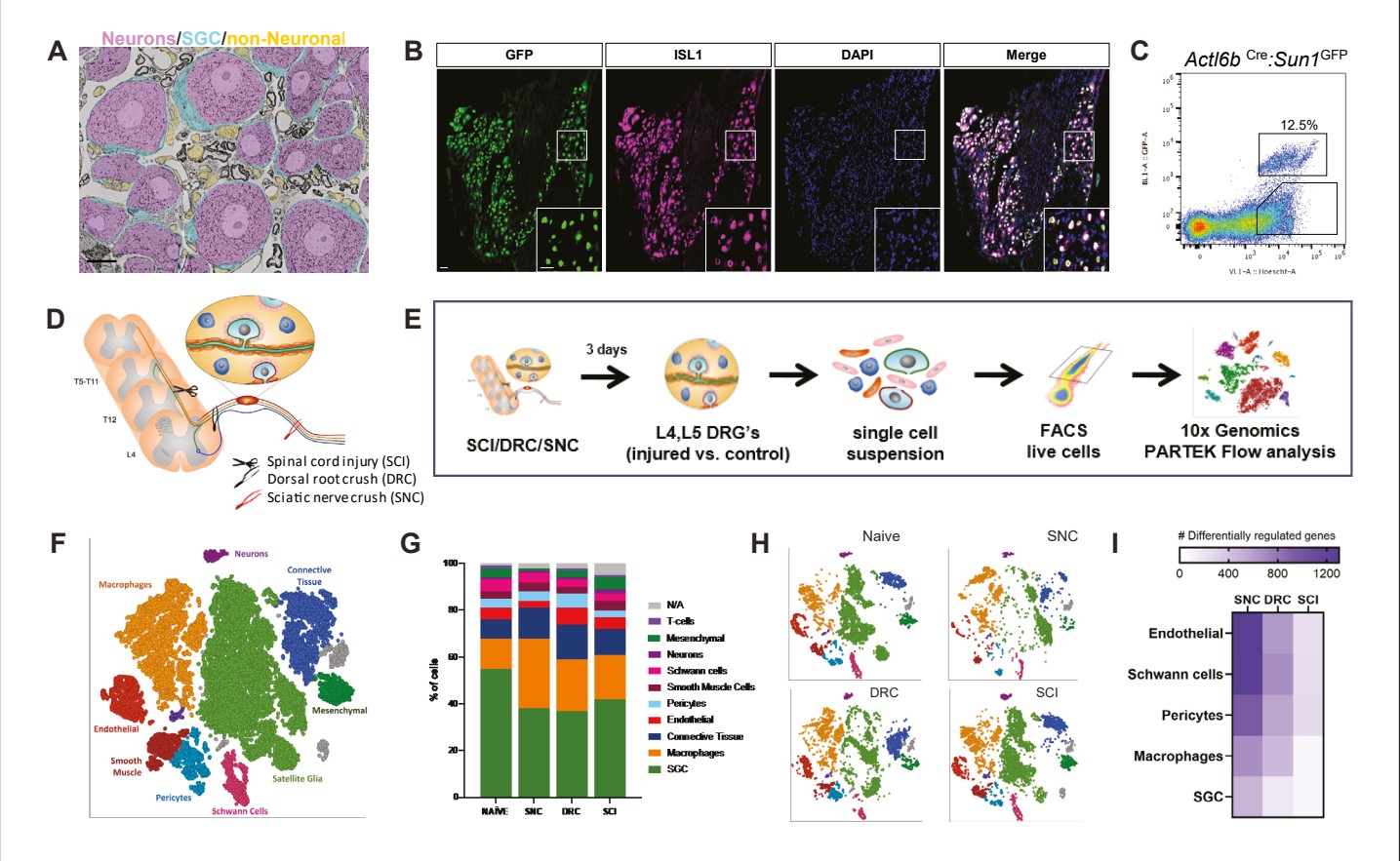

**Figure 1.** DRG cells respond differently following peripheral and central axon injuries. (**A**) Representative TEM images of a DRG section showing neuronal cell bodies (pseudo-colored in purple) its enveloping SGC (pseudo-colored in turquoise) and other non-neuronal cells (pseudo-colored in orange). n = 4 biologically independent animals. Scale bar:20 μm (**B**) *Actl6b*^Cre mice crossed with *Sun1*^GFP show expression of GFP in all neuronal cell somas, co-labeled with the unique neuronal marker ISL1 (magenta). n = 4 biologically independent animals, Scale bar: 50 μm (**C**) Flow cytometry analysis of dissociated DRG cells from *Actl6b*^Cre:*Sun1*^GFP mice. Scatter plot of fluorescence intensities of live Hoechst + cells (x axis) and GFP+ (y axis). 12.5 % of Hoechst + cells are also GFP+ positive. n = 3 biologically independent animals. (**D**) Diagram of mouse peripheral and central injury models. (**E**) Schematic of the experimental design for scRNAseq. (**F**) t-SNE plot of 25,154 cells from L4,L5 dissociated naïve and injured mouse DRG. 9 distinct cell clusters were assigned based on known marker genes. (**G**) Fraction of each cell type within naive (6343 cells), SNC (4735 cells), DRC (7199 cells) and SCI (7063 cells) conditions. n = 2 (NAI,DRC,SCI) and n = 1 (SNC) biologically independent experiments. (**H**) t-SNE plots of DRG cells separated by the different injury conditions, colored by cell type. (**I**) Heatmap of the number of differentially regulated genes in each cell type and injury condition (FDR ≤ 0.05, fold-change ≥ 2).

The online version of this article includes the following figure supplement(s) for figure 1:

**Source data 1.** Source files for scRNAseq analysis; top DEG for cell clustering and cluster counts.

**Source data 2.** Significant ligand-receptor interactions (p-value < 0.05).

**Figure supplement 1.** scRNAseq analysis of naïve and injured mouse DRG.

# Results

## Profiling sensory neuron microenvironment following peripheral and central injuries

Neurons are the largest cells in the DRG but are outnumbered by many non-neuronal cells (*Figure 1A*). FACS sorting analysis of dissociated DRG cells from *Actl6b*^Cre (*Baf53b*^Cre)*: Sun1*^GFP in which GFP is expressed in the neuronal nuclei (*Mo et al., 2015*; *Zhan et al., 2015*) showed that GFP-positive neurons represented only ~12.5 % of all cells (*Figure 1B and C*). To assess the DRG microenvironment response to central and peripheral axon injury, we performed single-cell RNA sequencing (scRNAseq) of L4, L5 mouse DRG 3 days after sciatic nerve crush injury (SNC), dorsal root crush injury (DRC), or spinal cord injury (SCI) using the Chromium Single Cell Gene Expression Solution (10 X Genomics)

(*Figure 1D and E*), as previously described (*Avraham et al., 2020*). Contralateral uninjured DRG were used as control and referred thereafter to naive. The sciatic nerve is composed of axons projecting from sensory neurons residing in multiple DRG, and SNC results in ~50 % of lumbar DRG neurons being axotomized (*Laedermann et al., 2014*; *Renthal et al., 2020*; *Rigaud et al., 2008*). SNC is followed by activation of a pro-regenerative program that allows functional recovery (*He and Jin, 2016*; *Mahar and Cavalli, 2018*). SCI injures the ascending axons of a subset of large diameter sensory neurons, leaving the descending axon branches in the spinal cord intact (*Attwell et al., 2018*; *Niu et al., 2013*; *Zheng et al., 2019*), and is not followed by regenerative outcomes. DRC damages all centrally projecting sensory axons in the PNS, without causing an impassable glial scar, and is followed by a slower regenerative growth compared to SNC that stops as axons reach the scar-free dorsal root entry zone (*Oblinger and Lasek, 1984*; *Smith et al., 2012*; *Wujek and Lasek, 1983*), providing an additional model to unravel the mechanisms promoting axon regeneration. The percent of DRG neurons lesioned under the three injury paradigms and the distance of the injury to the DRG may impact the injury responses of the microenvironment. However, all three injury paradigms are widely used models to study the mechanisms promoting axon regeneration.

Our scRNAseq protocol achieves efficient recovery of non-neuronal cells compared to other protocols that use single nuclear RNAseq to analyze neuronal responses to injury (*Avraham et al., 2020*; *Renthal et al., 2020*). While scRNAseq captures transcriptional responses, changes in RNA stability may also contribute to the differential profile, and the depth of sequencing obtained in scRNAseq analyses might not allow to capture low level transcripts. The number of total sequenced cells from all conditions was 25,154 from two biological replicates for naive, SCI and DRC conditions, and one biological replicate for SNC (*Figure 1—figure supplement 1A*), with an average of 45,000 reads per cell, 1500 genes per cell and a total of 17,879 genes detected (see filtering criteria in the methods). An unbiased (Graph-based) clustering, using Partek flow analysis package, identified 19 distinct cell clusters in the control and injured samples (*Figure 1—figure supplement 1B*). To identify cluster-specific genes, we calculated the expression difference of each gene between that cluster and the average in the rest of the clusters (ANOVA fold change threshold >1.5). Examination of the cluster-specific marker genes revealed major cellular subtypes including neurons (*Isl1*), SGC (*Fabp7*), endothelial cells *Pecam1*(Cd31), Schwann cells (*Ncmap*), pericytes *Kcnj8*(Kir6.1), smooth muscle cells (Pln), macrophages *Alf1*(Iba1), and connective tissue cells (*Col1a1*) (*Figure 1F*, *Figure 1—figure supplement 1D* and *Figure 1—source data 1*). A t-SNE (t-distributed stochastic neighbor embedding) plot of all 25,154 cells combined from naive and injury conditions revealed that SGC and macrophages clusters contained the largest number of cells (*Figure 1F*). Comparison of population distribution between the different injury conditions revealed a reduction in the percentage of SGC after peripheral and central injuries, with an increase in the number of macrophages compared to naive condition (*Figure 1G* and *Figure 1—source data 1*). Separate t-SNE plots for each condition uncovers major changes in cluster organization after SNC compared to naive, with less variations after DRC and high similarity between naive and SCI condition (*Figure 1H*, *Figure 1—figure supplement 1C*). We then determined the number of differentially expressed (DE) genes in endothelial cells, pericytes, Schwann cells, macrophages, and SGC (FDR ≤ 0.05, FC ≥2). Heat map of differential gene expression in the indicated cell types revealed that the magnitude of gene expression changes was the largest after SNC, but also occurred after DRC and SCI (*Figure 1I* and *Figure 1—source data 1*), as previously suggested (*Palmisano et al., 2019*; *Stam et al., 2007*).

To further investigate how the neuronal microenvironment is affected by the different injuries, we performed cell-cell interaction analysis based on ligand-receptor expression in the different cell types for every injury condition using CellPhoneDB repository (*Figure 1—source data 2*). This analysis revealed that the cell-cell interaction network changed significantly after SNC compared to naïve, and that these changes are distinct from those elicit by DRC. SCI had limited influence on the cellular network interaction compared to naive (*Figure 1—figure supplement 1E*). This analysis further highlights the importance of the microenvironment response and the potential extrinsic influence on axon regeneration.

## Alterations in blood-nerve-barrier markers in response to central and peripheral injuries

Blood-tissue barriers play an essential role in the maintenance and homeostasis of the tissue environment. Integrity of the peripheral nervous system is maintained by the blood-nerve-barrier (BNB), which shares many structural features with the blood brain barrier (*Richner et al., 2018*). An essential component of the BNB cellular architecture is tight junctions (TJ) in the endoneurial vascular endothelium or the perineurium that surrounds the nerve fascicle. Endothelial cells comprise the inner lining of vessels, while pericytes encompass blood microvessels such as blood capillaries (*Sims, 2000*). Sensory ganglia are highly vascularized (*Figure 2A*; *Jimenez-Andrade et al., 2008*), with blood vessels in sensory ganglia being more permeable than their counterpart in the brain (*Kiernan, 1996*; *Reinhold and Rittner, 2017*) or the nerve (*Hirakawa et al., 2004*; *Jimenez-Andrade et al., 2008*). Unlike in the brain, pericytes do not fully cover the blood vessel in peripheral nerve (*Stierli et al., 2018*). We observed a similar situation in the DRG, with the presence of blood vessel not fully covered by pericytes (*Figure 2B*). We examined changes in gene expression that occurred in endothelial cells and pericytes following peripheral and central injuries (FDR ≤ 0.05, FC ≥2) (*Figure 2—source data 1*), as the magnitude of gene expression changes was the largest in these cells after SNC (*Figure 1I*). t-SNE plots of endothelial cells and pericytes demonstrated different clustering of cells after SNC or DRC, while similar clustering in naïve and after SCI were observed (*Figure 2C and D*). Increased BNB permeability in the nerve is linked to changes in the expression of TJ genes, in particular a reduced expression of ZO-1 (*Tjp1*) in endoneurial cells (*Richner et al., 2018*). We thus examined the expression of tight junction (TJ) as well as adherens junction (AJ) genes. Heat map of TJ and AJ genes indicated that the response of barrier components was affected by SNC differently than DRC, with numerous junction genes being differentially expressed following SNC and DRC compared to naive and SCI condition (*Figure 2E and F*). Changes in *Tjp1* and *Tjp2* expression suggest that the BNB may be more permeable after SNC and DRC compared to naive and SCI. KEGG pathway analysis of DE genes in endothelial cells and pericytes further suggest that the BNB may be differentially altered after SNC and DRC (*Figure 2G and H*). The enrichment of the cell cycle pathway after SNC and DRC suggests that endothelial cell division may regulate blood vessel angiogenesis (*Zeng et al., 2007*).

After nerve injury, dedifferentiation of Schwann cells into repair Schwann cells at the site of injury as well as resident macrophages in the nerve elicits breakdown of the BNB (*Mellick and Cavanagh, 1968*; *Napoli et al., 2012*; *Richner et al., 2018*). Although Schwann cells in the DRG are far away from the injury site in axons, we found that they undergo transcriptional changes that are distinct after peripheral and central injuries. t-SNE plots demonstrated different clustering of Schwann cells in SNC and DRC conditions, with similar clustering in naïve and SCI (*Figure 2I*). We next examined the expression of genes known to promote differentiation of Schwann cells into repair Schwann cells (*Jessen and Arthur-Farraj, 2019*). Heat map of such genes revealed some non-overlapping changes after all three injuries (*Figure 2J* and *Figure 2—source data 1*). Notably, *Ngf*, which is known to promote myelination by Schwann cells in peripheral nerves (*Chan et al., 2004*) is downregulated after all injuries. VEGF is known to increase BNB permeability (*Lim et al., 2014*) and *Vegfb* is differentially regulated after peripheral and central injuries (*Figure 2J*), suggesting that Schwann cells may influence BNB permeability in the DRG. *Shh is* strongly upregulated after DRC. Shh signaling in Schwann cells in the DRG after SNC and DRC may have neuroprotective functions (*Hashimoto et al., 2008*) and may facilitate axon regeneration (*Martinez et al., 2015*). KEGG analysis also revealed that the hedgehog signaling pathway and axon guidance is upregulated specifically after DRC (*Figure 2K*). The hippo signaling pathway, which plays multiple cellular functions, such as proliferation, apoptosis, regeneration, and organ size control (*Yu and Guan, 2013*; *Zhao et al., 2011*), is downregulated specifically after SCI and DRC. Key transcription factors in the Hippo pathway, *Yap* and *Bmp5* were downregulated after DRC and SCI, and upregulated after SNC (FDR ≤ 0.05, FC ≥2) (*Figure 2—source data 1*). These results suggest that Schwann cell in the DRG respond differently to peripheral and central injuries, with central injury potentially limiting their plasticity.

## Macrophages proliferate in response to peripheral but not central axon injuries

After nerve injury, breakdown of the BNB allows the influx of inflammatory cells at the site of injury in the nerve to promote repair (*Mellick and Cavanagh, 1968*; *Napoli et al., 2012*). In addition to their

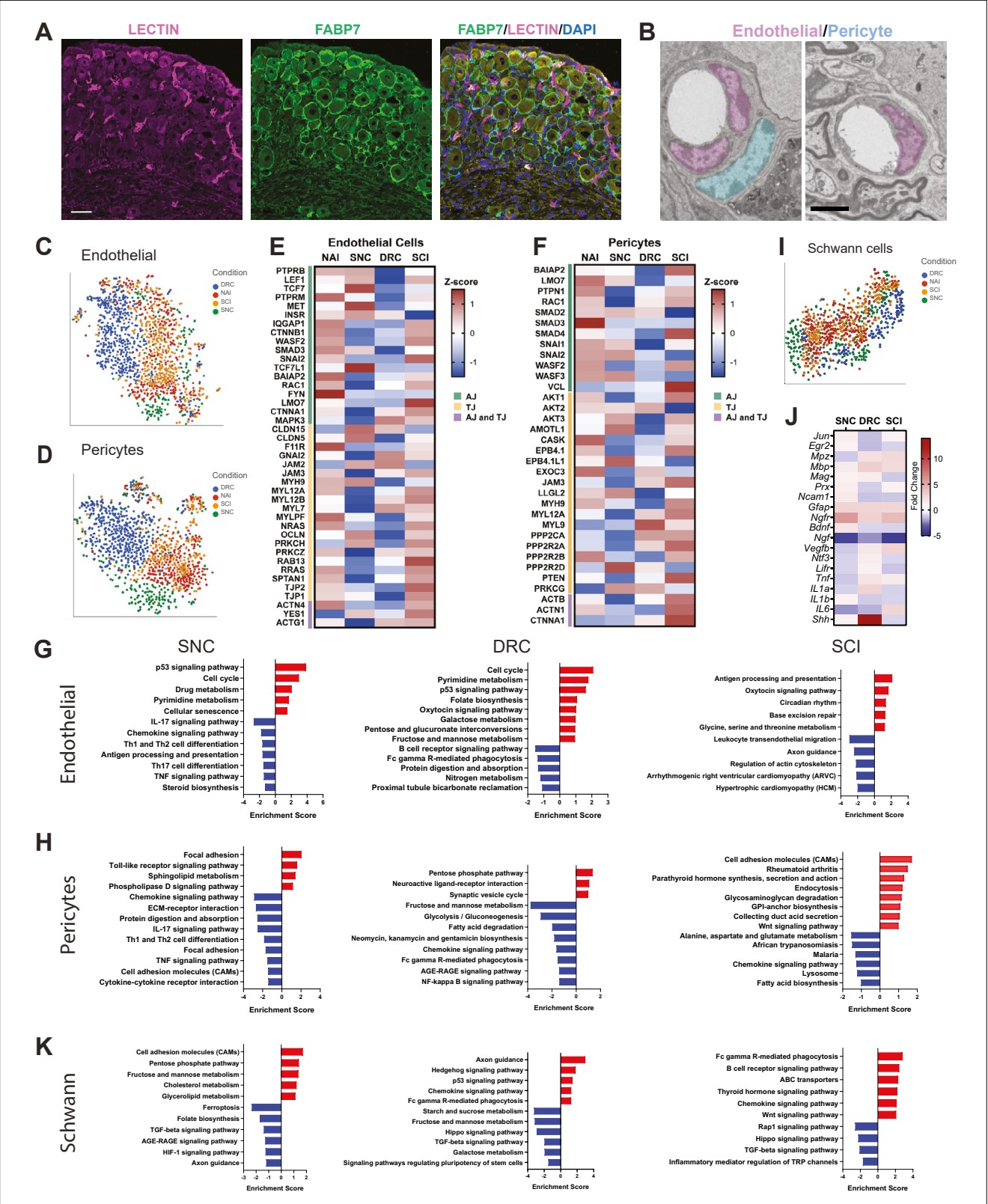

**Figure 2.** Molecular changes in non-neuronal cells in response to central and peripheral injuries. (**A**) Representative images of mouse DRG sections injected with Lycopersicon esculentum (Tomato) Lectin (magenta), labeling blood vessels and immunostained with FABP7 (green) labeling SGC. Scale bar: 50 µm. (**B**) Representative TEM images of DRG sections focusing on blood vessels with the surrounding endothelial (pseudo-colored in purple) and pericytes (pseudo-colored in turquoise) n = 4 biologically independent animals. Scale bar:2 µm (**C**) t-SNE plot of DRG endothelial cells colored by injury

*Figure 2 continued on next page*

**Figure 2 continued**

condition. (**D**) t-SNE plot of DRG pericytes colored by injury condition. (**E**) Heatmap of Adherens junction (AJ) and Tight Junction (TJ) genes expression in endothelial cells by z-score for all injury conditions. (**F**) Heatmap of Adherens junction (AJ) and Tight Junction (TJ) genes expression in pericytes by z-score for all injury conditions. (**G**) Pathway analysis (KEGG 2019) of differentially upregulated (red) and downregulated (blue) genes in the endothelial cell cluster. n = 2 biologically independent experiments. (FDR ≤ 0.05, fold-change ≥2). (**H**) Pathway analysis (KEGG 2019) of differentially upregulated (red) and downregulated (blue) genes in the pericyte cluster. n = 2 biologically independent experiments. (FDR ≤ 0.05, fold-change ≥ 2). (**I**) t-SNE plot of DRG Schwann cells colored by injury condition. (**J**) Heatmap of fold change expression for selected repair Schwann cell genes after SNC, DRC and SCI compared to naïve. (**K**) Pathway analysis (KEGG 2019) of differentially upregulated (red) and downregulated (blue) genes in the Schwann cell cluster. n = 2 biologically independent experiments (FDR ≤ 0.05, fold-change ≥ 2).

The online version of this article includes the following figure supplement(s) for figure 2:

**Source data 1.** Source files for scRNAseq analysis; DEG in endothelial cells, pericytes and Schwann cells in response to peripheral and central injuries (FDR ≤ 0.05, fold-change ≥ 2), repair Schwann cell genes expression across injuries.

role at the site of injury in the nerve, macrophages regulate axon regeneration and pain responses acting at the level of the ganglia (*Kwon et al., 2013*; *Niemi et al., 2013*; *Yu et al., 2020*). Both resident and infiltrating macrophages were found in the DRG (*Zigmond and Echevarria, 2019*). To understand if macrophages in the DRG include the two major macrophages subsets found in the nerve, snMac1 that reside in the endoneurium or snMac2 that reside in the connective tissue surrounding nerve fascicles (*Ydens et al., 2020*), we analyzed the percent of DRG macrophages expressing marker genes for these two subtypes. This analysis revealed that most DRG macrophages express the snMac1 genes (*Cbrr2*, *Mgl2*) (*Figure 3A*, light blue), whereas few DRG macrophages express the snMac2 genes (*Retnnla*, *Clecl10a*, *Folr2*) (*Figure 3A*, blue). The DRG macrophages, similarly to nerve macrophages (*Wang et al., 2020*; *Ydens et al., 2020*), also express CNS associated microglia genes such as *Tmem119*, *P2ry12*, and *Trem2* (*Figure 3A*, pink) and common microglia/macrophages markers *Ccl12*,*Gpr34*, *Gpr183*, *Hexb*, *Mef2c*, *St3gal6*, and *Tagap* (*Figure 3A*, green). The common macrophages markers *Cd68*, *Emr1* and *Aif1* were expressed in >80% of cells in the macrophage cluster (*Figure 3A*, orange and *Figure 3—source data 1*). These results suggest that DRG macrophages share similar properties to snMac1 residing in the nerve endoneurium and with CNS microglia.

We next examined the injury responses of DRG macrophages. The number of macrophages increased after SNC compared to naïve and also increased to a lesser extent after DRC and SCI (*Figure 1G*). Macrophages displayed a similar gene expression profile in naive and SCI condition, but SNC and DRC elicited large changes in genes expression (*Figure 3B*). KEGG pathway analysis of DE genes (FDR ≤ 0.05, FC ≥2) (*Figure 3—source data 1*) revealed upregulation of cell cycle and DNA replication after SNC, while DRC and SCI macrophages mainly showed upregulation of metabolic pathways such as steroid biosynthesis and glycolysis/gluconeogenesis pathways (*Figure 3C*). Interestingly, macrophages from all injury conditions down regulated genes related to antigen processing and presentation as well as genes involved in phagosome activity (*Figure 3C*). We further validated the down regulation of genes involved in antigen processing and presentation associated with class II major histocompatibility complex (MHC II) *Cd74*, *H2-Aa* and *Ctss* in qPCR experiments (*Figure 3D* and *Figure 3—source data 1*). Several studies suggested a predominant anti-inflammatory macrophage phenotype in the DRG following sciatic nerve injury (*Komori et al., 2011*; *Kwon et al., 2013*; *Lindborg et al., 2018*; *Niemi et al., 2013*). Heat map of cytokines and other macrophages polarization markers suggest that macrophages responses to injury in the DRG are complex and may not easily follow the classical pro-inflammatory and anti-inflammatory polarization scheme (*Figure 3E* and *Figure 3—source data 1*). Furthermore, expression of the anti-inflammatory marker gene *Arg1* was only detected after DRC and the pro-inflammatory marker *Nos2* was not detected in any conditions (*Figure 3E*). Among the significant down regulated genes after SNC, we found the cytokines *Ccl2*, *Il1b* and *Tnf*, which we validated for their downregulation by qPCR experiments (*Figure 3D* and *Figure 3—source data 1*). Hematopoietic cell lineage pathway, which is involved in the formation of macrophages from myeloid cells, was down regulated specifically after DRC and SCI, but not after SNC (*Figure 3C*). We next explored cell division in the macrophage cluster. The cell cycle and DNA replication pathways were upregulated only after SNC (*Figure 3C*). Heatmap analysis of the proliferation markers *Mki67*, *Cdk1* and *Top2a* further demonstrated higher expression of proliferation genes after SNC compared to naive macrophages and following central injuries (*Figure 3E*). qPCR analysis of DRG cells after SNC showed increase in *Mki67* expression compared to naive (*Figure 4D*). t-SNE

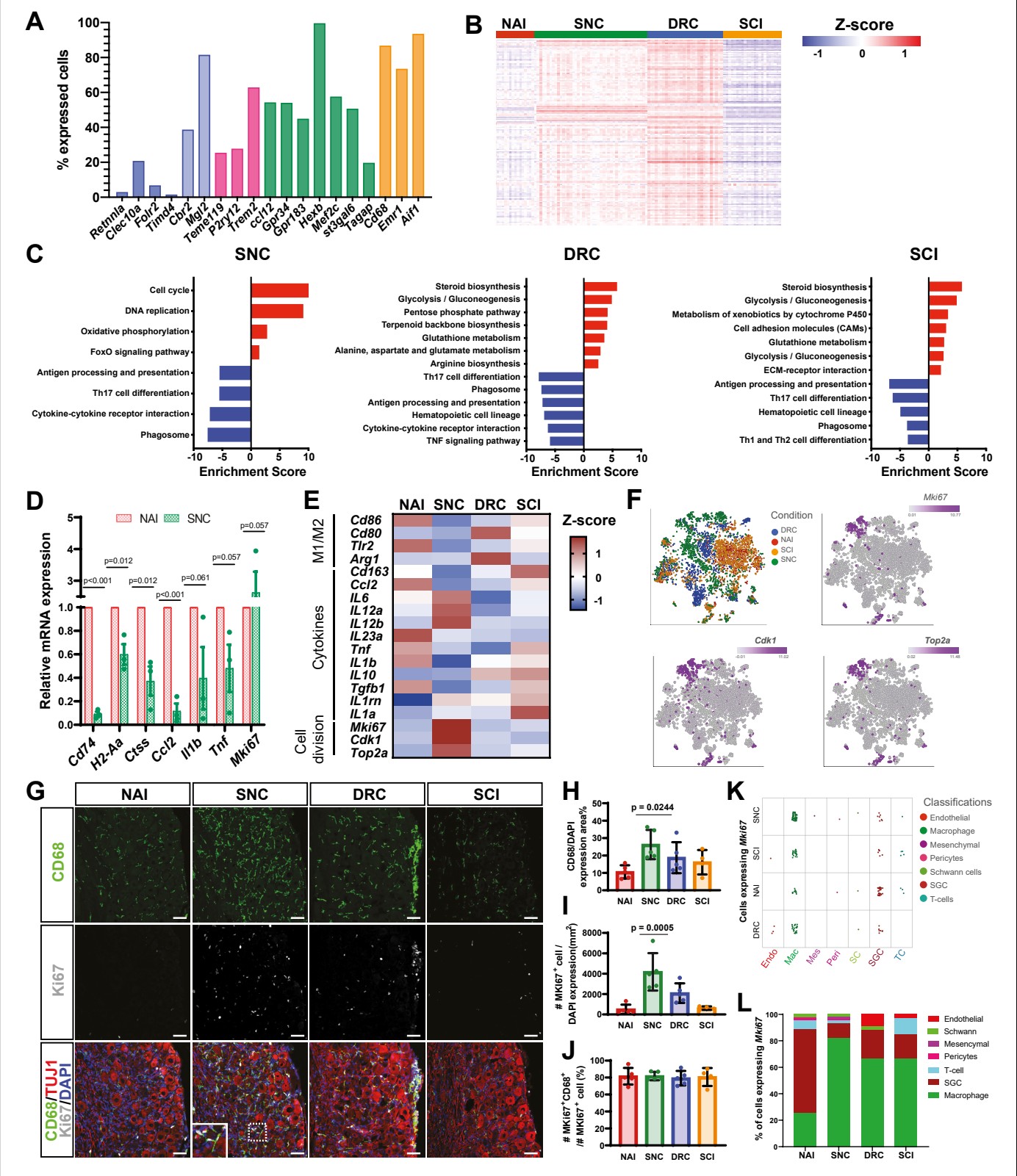

**Figure 3.** Macrophages undergo distinct transcriptional changes in response to central and peripheral injuries. (**A**) Fraction of uninjured cells expressing selected genes in the macrophage cluster. snMac2 (blue), snMac1 (light blue), specific microglia genes (pink), CNS microglia/macrophages (green) and common macrophage markers (orange). (**B**) Heatmap of gene expression profile in macrophages by z-score for all injury conditions. (**C**) Pathway analysis (KEGG 2019) of differentially upregulated (red) and downregulated (blue) genes in the Macrophage cell cluster. n = 2 biologically independent

*Figure 3 continued on next page*

*Figure 3 continued*

experiments. (FDR ≤ 0.05, fold-change ≥ 2). (**D**) DRG qPCR analysis of DEG in macrophages after SNC compared to Naive. (**E**) Heatmap of M1, M2 macrophage markers, selected cytokines and proliferation marker gene expression by z- score for all injury conditions. (**F**) t-SNE plots of mouse DRG macrophages colored by injury condition and t-SNE overlay for expression of proliferation marker genes in pooled macrophage cluster from all injury conditions. (**G**) Representative images of immunofluorescence staining of DRG sections labeled with CD68 (green), MKI67 (white) and TUJ1 (red) from naïve mice, SNC, DRC and SCI injuries n = 5 biologically independent animals. Scale bar: 50 μm. (**H**) Quantification of area with CD68 expressing cells. (**I**) Quantification of MKI67 expressing cells normalized to DAPI. (**J**) Quantification of the percentage of cells expressing both MKI67 and CD68 out of all MKI67 positive cells. n = 5 (NAI,SNC,DRC) and n = 4 (SCI) biologically independent animals. (**H-J**) One-way analysis of variance (ANOVA) followed by Bonferroni's multiple comparisons test. Data are presented as mean values ± SD. (**K**) Plot of cells expressing *Mki67*, colored by cell type, for all injury conditions. Every dot represents one cell. (6< log gene counts). (**L**) Quantification of the percentage of cells expressing *Mki67*, colored by cell type, in all injury conditions.

The online version of this article includes the following figure supplement(s) for figure 3:

**Source data 1.** Source files for scRNAseq analysis; Macrophage markers in the macrophage cluster, DEG in macrophages (FDR ≤ 0.05, fold-change ≥ 2) and *MkKi67* expression.

plots overlaid with the proliferation marker genes *Mki67*, *Cdk1* and *Top2a* revealed expression mainly in one macrophage subtypes following SNC (~60%) and smaller clusters of proliferating cells after DRC (~20%), SCI (~14%) and in naive condition (~6%) (*Figure 3F*). Validation of the scRNAseq data by immunostaining of DRG sections with the macrophage-specific marker CD68 and the proliferation marker MKI67 further revealed a higher number of CD68 and MKI67-positive cells after SNC and a trend toward an increase after DRC (*Figure 3G–I* and *Figure 3—source data 1*). Higher magnification in sections demonstrates co-expression of MKI67 in CD68-positive cells (*Figure 3G*). Image quantification across all conditions demonstrated that 80 % of the MKI67 positive cells were also positive for CD68 (*Figure 3J*). Analysis of *Mki67* expressing cells in the scRNAseq data revealed a majority of cells in the macrophage cluster, with highest abundance after SNC (82 % of cells), DRC and SCI (67 % of cells) (*Figure 3K and L* and *Figure 3—source data 1*). These results suggest that macrophages represent a large proportion of proliferating cells in the DRG after nerve injury. This is consistent with the recent observation that macrophage expansion after nerve injury in the DRG involves proliferation (*Yu et al., 2020*). However, whether the proliferating macrophages originate from resident macrophages or from the infiltration of monocytes-derived macrophages remains to be determined. These results highlight that central and peripheral nerve injury differently affect gene expression in macrophages and that a better understanding of these responses may highlight their role in pain and nerve regeneration.

## A subset of macrophages expressing glial markers is increased by injury

The macrophage cluster was classified in our scRNAseq analysis by differential expression of macrophage specific markers such as *Cd68* and *Aif1* (*Figure 1—figure supplement 1D*). However, a subcluster with 484 cells was classified as cluster 13 in an unbiased clustering (*Figure 1—figure supplement 1B*). Examination of expression of top marker genes in macrophage (*Cd68*) and SGC (*Fabp7*) revealed co-expression of both markers in cluster 13 (*Figure 4A*, red circle). Macrophages (orange cells) and SGC (green cells) clusters were then specifically plotted for expression of *Cd68* and *Fabp7*, demonstrating that a subset of cells co-express macrophage and SGC markers (*Figure 4B*). This agrees with other studies reporting that SGC can express immune markers (*Donegan et al., 2013*; *Huang et al., 2021*; *Jasmin et al., 2010*; *Mapps et al., 2021*; *van Velzen et al., 2009*; *van Weperen et al., 2021*). Violin plots for *Cd68* and *Fabp7* expression across all cell types in the DRG further demonstrate that *Cd68* and *Fabp7* are highly expressed in cluster 13, which we named the 'Imoonglia' cluster (*Figure 4—figure supplement 1A*). Violin plot of total counts in all cell clusters excluded the possibility that this cluster represents doublets or SGC phagocytosed by macrophages (*Figure 4—figure supplement 1B*). Dot plot analysis for expression of the macrophage marker genes *Aif1*, *Cd68* and *Cx3cr1* and the glia marker genes *Fabp7*, *Cadh19*, and *Plp1*, further support co-expression of both macrophage and SGC markers in the Imoonglia cluster (*Figure 4C* and *Figure 4—source data 1*). A trajectory analysis further demonstrates that Imoonglia express a transcriptome that position them between SGC and macrophages (*Figure 4D*). To validate the scRNAseq results, which showed expression of macrophage/myeloid markers in the Imoonglia cells (*Figure 4—figure*

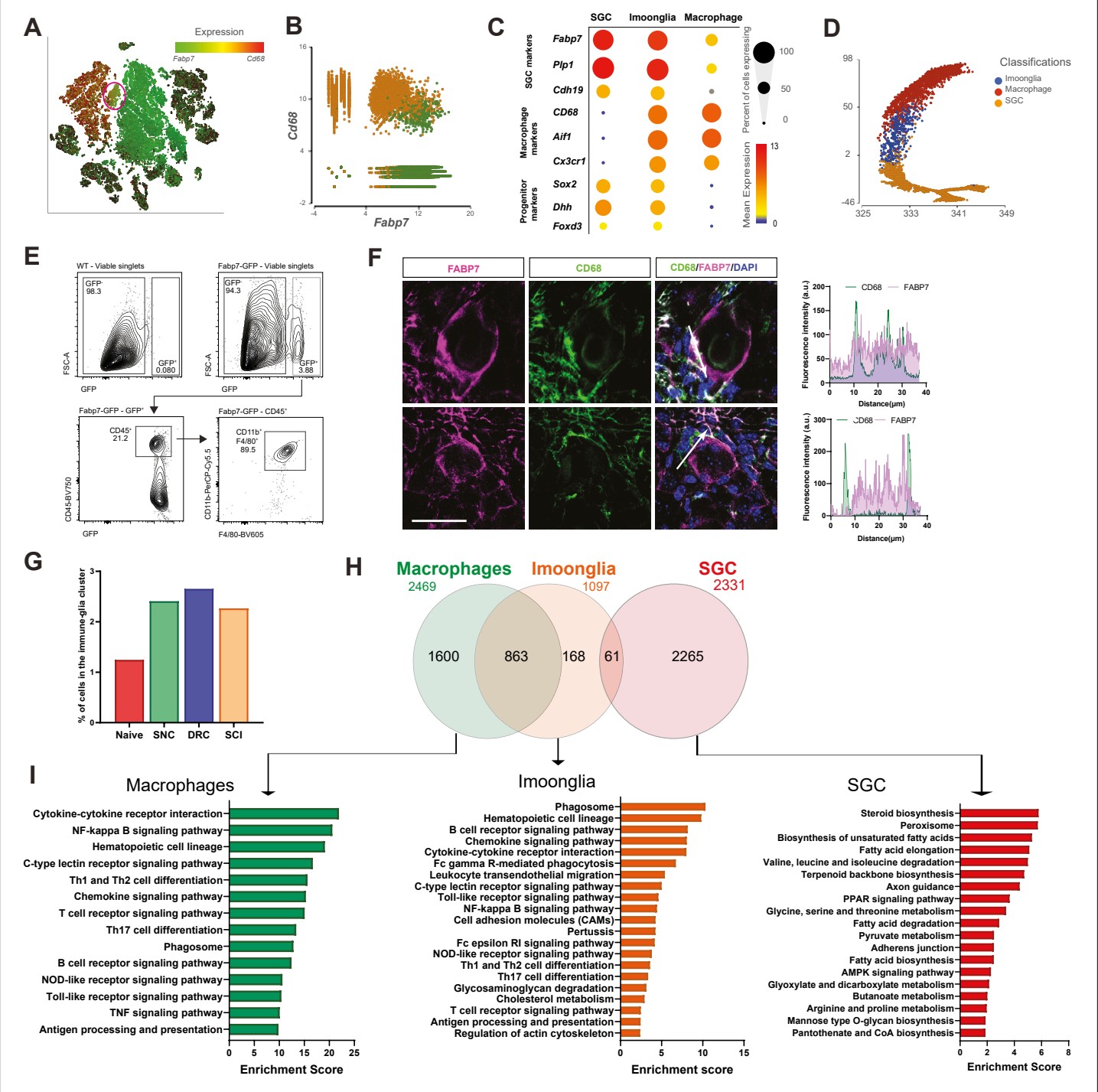

**Figure 4.** A subset of macrophages expressing glial markers is increased by injury. (**A**) t-SNE plot of cells from all injury conditions overlay for expression of *Cd68* (red) and *Fabp7* (green). (**B**) Plotting cells in the macrophage (orange) and SGC (green) clusters for expression of *Cd68* (y-axis) and *Fabp7* (x-axis). (**C**) Dot plot of macrophage, glial and progenitor marker genes expression in the macrophage, SGC and Imoonglia clusters. The percentage of cell expressing the gene is calculated as the number of cells in each cluster express the gene ( > 0 counts) divided by the total number of cells in the respective cluster. Expression in each cluster is calculated as mean expression of the gene relative to the highest mean expression of that gene across all clusters. (**D**) Trajectory analysis of macrophage, SGC and Imoonglia cell clusters. (**E**) Flow cytometry analysis of DRG cells from *Fabp7*creER:*Sun1*GFP mice, stained with the macrophage marker genes CD45, CD11b and F4/80 n = 2 (F4/80) n = 1 (CD45,CD11b,F4/80) biologically independent animals. (**F**) Representative confocal images of immunofluorescence staining of DRG sections labeled with CD68 (green) and FABP7 (magenta). Fluorescence intensity for CD68 and FABP7 was measured along the arrow line. Scale bar: 50 µm (**G**) Fraction of cells in the Imoonglia cluster by injury condition. n = 2 biologically independent experiments. (**H**) Venn diagram of differentially expressed genes in the Imoonglia cluster (1,097 genes) was compared to

*Figure 4 continued on next page*

*Figure 4 continued*

top differentially expressed genes in the macrophage cluster (2,469 genes) and the SGC (2,331 genes) (FDR ≤ 0.05, fold-change ≥ 2). (I) Pathway analysis (KEGG 2019) of differentially expressed genes in Macrophages, Imoonglia and SGC.

The online version of this article includes the following figure supplement(s) for figure 4:

**Source data 1.** Source files for scRNAseq analysis, flow cytometry additional experiments.

**Figure supplement 1.** Imoonglia- a subset of macrophage expressing glia markers.

*supplement 1E*), we performed a flow cytometry experiment. We found that a subset of genetically labeled SGC (*Fabp7*^creER:*Sun1*^GFP) (*Avraham et al., 2020*) express the specific macrophage/myeloid markers CD11B, F4/80 and CD45 (*Figure 4E* and *Figure 4—source data 1*). To further confirm the presence of this cell population, we performed co-immunostaining of DRG sections with CD68 and FABP7. We observed co-expression of both markers in a small population of cells with SGC morphology surrounding sensory neurons (*Figure 4F* upper, *Video 1*). Macrophages can also be located in close proximity to SGC (*Figure 4F* bottom), suggesting that their localization around sensory neurons can resemble SGC (*Avraham et al., 2020*; *Hanani, 2005*). Examination of the extent of the Imoonglia population in the DRG revealed that this is a rare population, representing ~1 % of all DRG cells (*Figure 4G*). Interestingly, the representation of Imoonglia in the DRG increased to ~2.5 % after both peripheral and central injuries (*Figure 4G*). t-SNE plot of Imoonglia cells revealed similar clustering across all conditions, suggesting similar gene expression that is not affected by injury (*Figure 4—figure supplement 1C*). We then pooled the Imoonglia cluster from all conditions and calculated the expression difference of each gene between that cluster and the average in the rest of the clusters ( > 1.5 -fold change p-value < 0.05). We then compared the genes uniquely expressed in Imoonglia (1097 genes) to macrophages (2469 genes) and SGC (2331 genes) (*Figure 4—source data 1*). This analysis revealed a higher similarity of Imoonglia cluster to macrophages (863 shared genes) than SGC (61 shared genes) (*Figure 4H*, *Figure 4—source data 1*). Imoonglia cells express many of the known macrophage/myeloid markers, *Cd45*, *Cd206*, *Cd163*, *Cd14*, *Cd209*, and *Cd38* (*Figure 4—figure supplement 1E* and *Figure 4—source data 1*), and, interestingly, also some of the glial linage progenitor markers *Sox2*, *Dhh,* and *Foxd3* (*Figure 4C*). KEGG pathway analysis of unique Imoonglia genes (168) and macrophage genes (1600) reveals similarity in immune-related pathways such as antigen processing and presentation, phagosome and hematopoietic cell lineage, while the top expressed pathways in SGC (2265 genes) relate to steroid biosynthesis, peroxisome and fatty acid metabolism (*Figure 4I*). KEGG analysis of the shared Imoonglia/Macrophage genes (863) mainly represented immune pathways while the shared Imoonglia/SGC genes (61) was enriched for metabolic pathways and cell adhesion (*Figure 4—figure supplement 1D* and *Figure 4—source data 1*). Together, our analysis supports the existence of cells with SGC morphology that express immune markers and provides a comprehensive characterization of this rare Imoonglia cell type.

## SGC represent a diverse cell population

We next determined if SGC represent a diverse glial population in the DRG in naive conditions. We previously described that *Fabp7* is a specific marker gene for SGC in DRG and that the FABP7 protein is expressed in all SGC in the DRG (*Figure 5A*; *Avraham et al., 2020*). The DRG encompasses different types of sensory neurons such as nociceptors, mechanoreceptors, and proprioceptors (*Renthal et al., 2020*; *Usoskin et al., 2015*), with each type controlling a different sensory function. To determine if SGC also exist as different subtypes in DRG, we examined the expression of other known SGC markers, *Cadh19*, *Kcnj10* and *Glul (GS)* in addition to *Fabp7*, by pooling naive SGC from our scRNAseq analysis. *Fabp7* was expressed in over 90 % of SGC, whereas *Cadh19*, *Kcnj10* and *Glul* expressed only in ~50 % of SGC (*Figure 5B and C* and *Figure 5—source data 1*). An unbiased clustering of SGC from naive DRG revealed four different subtypes with distribution

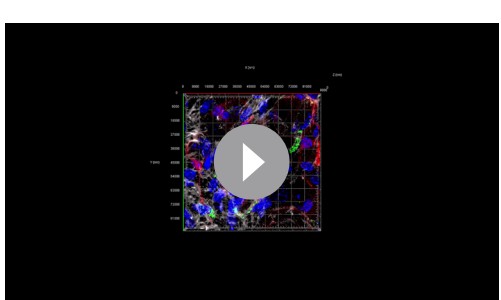

**Video 1.** 3D video of Imoonglia Glia.
https://elifesciences.org/articles/68457/figures#video1

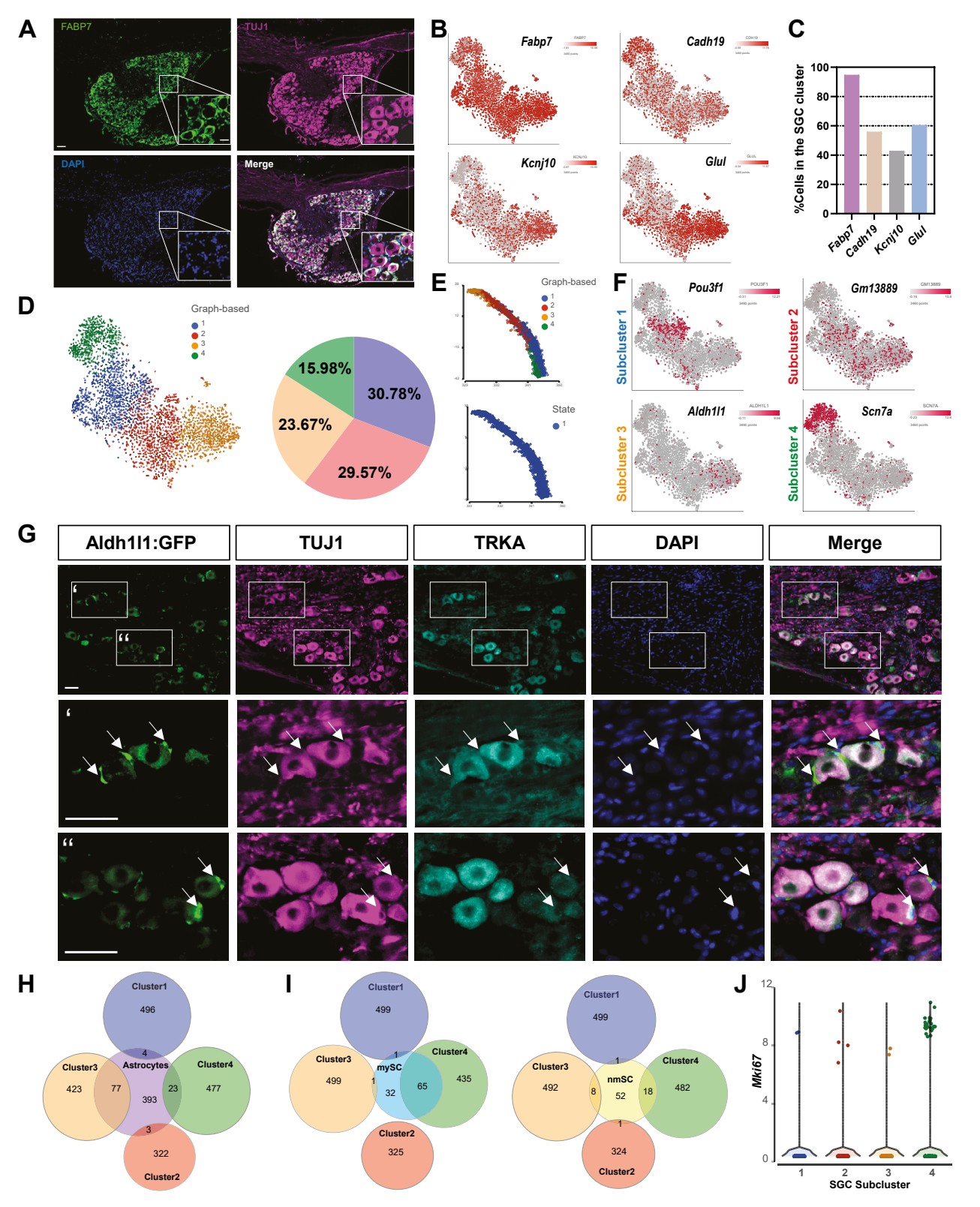

**Figure 5.** SGC represent a diverse cell population. (**A**) Representative images of immunofluorescence staining of DRG sections labeled with FABP7 (green) and TUJ1 (magenta). n = 4 biologically independent animals. Scale bar: 100 µm, zoomed image: 50 µm (**B**) t-SNE overlay for expression of SGC marker genes in pooled SGC cluster from naïve mice. (**C**) Fraction of cells in the SGC cluster expressing the SGC marker genes *Fabp7*, *Cadh19*, *Kcnj10,* and *Glul*. (6< log gene counts). (**D**) t-SNE plot of SGC cluster colored by subclusters (unbiased, Graph based clustering) with quantification

*Figure 5 continued on next page*

*Figure 5 continued*

of the fraction of cells in the different SGC subclusters out of total number of naïve SGC. (**E**) Trajectory analysis of SGC subclusters. (**F**) t-SNE overlay for expression of top differentially expressed genes in SGC subclusters. (**G**) Representative images of immunofluorescence staining of DRG sections from Aldh1l1::Rpl10a-Egfp mice (green) labeled with TUJ1 (magenta) and TRKA (cyan). n = 4 biologically independent animals. Scale bar 50 μm (**H**) Venn diagram comparing signature genes in SGC subclusters and astrocytes. (**I**) Venn diagrams comparing signature genes in SGC subclusters with myelinating (mySC) and non-myelinating (nmSC) Schwann cells markers. (**J**) Violin plot for expression of *Mki67* across SGC subclusters.

The online version of this article includes the following figure supplement(s) for figure 5:

**Source data 1.** Source files for scRNAseq analysis; SGC marker genes expression, DEG in SGC subclusters (FDR ≤ 0.05, fold-change ≥2), Astrocytes and Schwann cells.

**Figure supplement 1.** Unique enriched pathways of SGC subclusters.

between 15% and 30% for each subcluster that are represented by unique sets of gene expression (*Figure 5D* and *Figure 5—source data 1*). Trajectory analysis of SGC subtypes indicates a path starting from cluster 3, to cluster 2, then cluster 1 and finally cluster 4, with the same transcriptional state in all subtypes (*Figure 5E*). Overlay of *Pouf3f1*, *Gm13889*, *Aldh1l1*, and *Scn7a* in t-SNE plots demonstrated cluster specific expression (*Figure 5F*). KEGG pathway analysis of each sub-cluster highlights distinct functions, with cluster 1 enriched for glycan biosynthesis and MAPK signaling, cluster 2 enriched for cytokine and IL-17 signaling, cluster 3 enriched for steroid biosynthesis and terpenoid backbone biosynthesis and cluster 4 enriched for ECM and cell adhesion pathways (*Figure 5—figure supplement 1*).

To determine if a given subtype is associated with a specific neuronal subtype, we examined expression of cluster 3 specific gene *Aldh1l1* using the Aldh1l1::Rpl10a-Egfp reporter mouse (*Doyle et al., 2008*). Although *Scn7a* appeared to be the most cluster-specific gene marker, it is also expressed in neurons, impairing the precise examination of its cellular localization. Aldh1l1 is typically used to label all astrocytes in the CNS and we found that *Aldh1l1* drove expression of Rpl10a-Egfp in a subset of SGC, consistent with the single cell data (*Figure 5G*). Rpl10a-Egfp expression was detected in SGC surrounding both TRKA positive nociceptor neurons and TRKA negative neurons (*Figure 5G*), suggesting that cluster 3 SGC are not specifically associated to a given neuronal subtype. Our results are also consistent with the recent finding that Aldh1l1::Rpl10a-Egfp mice express Egfp in a subset of SGC (*Rabah et al., 2020*). We next compared each SGC subcluster with astrocytes (*Zhang et al., 2014*), myelinating Schwann cells and non-myelinating Schwann cells (*Wolbert et al., 2020*). Cluster 3, which expresses the astrocyte marker *Aldh1l1* shares the most genes with astrocytes such as *Glul*, *Kcnj10,* and *Slc1a3* (Glast) (*Figure 5H*, *Figure 5—figure supplement 1*). Cluster 4, which is highly enriched for *Scn7a* (*Figure 5F*), shares more similarities with myelinated Schwann cells, consistent with the expression of *Scn7a* in myelinating Schwann cells (*Watanabe et al., 2002*; *Figure 5I*). Clusters 1 and 2 represent the most unique SGC subtypes (*Figure 5H,I*). Interestingly, some SGC express the proliferation marker *Mki67* in naive DRGs (*Figure 3K and L* and *Figure 3—source data 1*), with specific enrichment in cluster 4 (*Figure 5J*). Whether the SGC subtypes represent functionally distinct populations remains to be determined.

## A distinct SGC cluster appears in response to peripheral nerve injury

We previously revealed the contribution of SGC to axon regeneration (*Avraham et al., 2020*). To determine if the different regenerative capacities after peripheral or central injury result in part from different responses in SGC, we examined the SGC responses to SCI and DRC compared to SNC. Separate clusters emerged in SGC after SNC and DRC injuries but were similar in naïve and SCI conditions (*Figure 6A*). We next determined if the four sub clusters identified in naive conditions (*Figure 5D*) are changing following the different injuries. An unbiased clustering of SGC in all conditions recognized seven different sub clusters, in which clusters 1–4 represent the four clusters found in naive conditions (*Figure 6B*, *Figure 6—source data 1* and *Figure 5—source data 1*). The percentage of cells in clusters 2 and 5 remained largely unchanged after the different injuries, whereas the percentage of cells in cluster 1, 4, 6, and 7 were up regulated by injury conditions (*Figure 6C*). After SNC, cluster 1 (blue) decreased, whereas cluster 6 (light blue) emerged and accounted for 40 % of all SGC. In contrast, SGC after SCI showed a decrease in cluster 4 (green), with cluster 7 appearing specifically after SCI (pink). Dot plot analysis further supports sub-cluster changes in SGC following different injuries, revealing the percentage of cells in each cluster together with the level of expression of

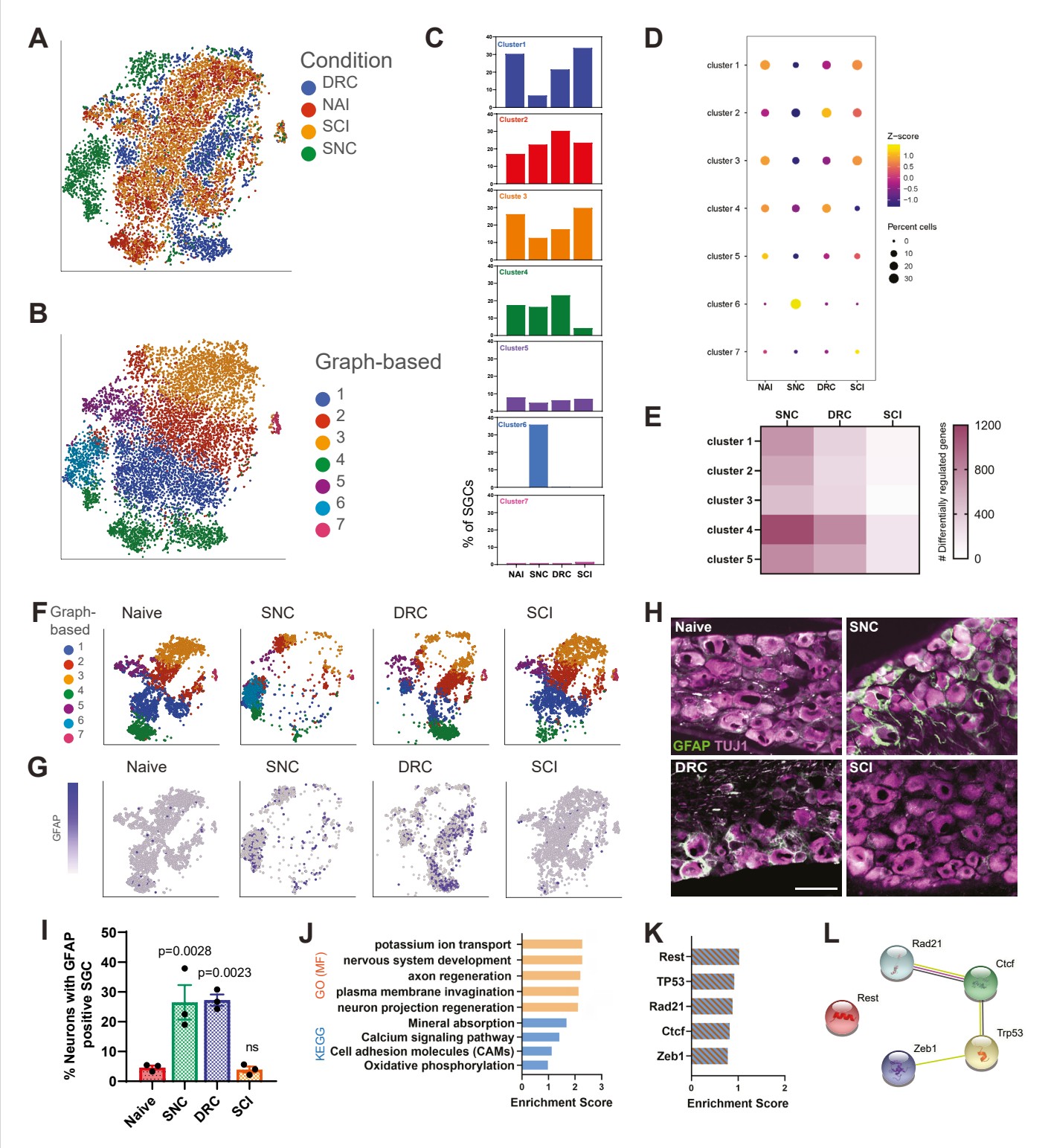

**Figure 6.** A distinct SGC cluster appears in response to peripheral nerve injury. (**A**) t-SNE plot of pooled SGC from naïve and injured mice, colored by injury condition. (**B**) t-SNE plot of pooled SGC from naïve and injured mice, colored by unbiased clustering. (**C**) Quantification of the fraction of cells for each subcluster in the different injury conditions. (**D**) Dot plot of SGC subclusters representation in the different injury conditions by z-score. The percentage of cells in a subcluster is divided by the total number of cells in the respective condition. (**E**) Heatmap of the number of differentially regulated genes in each SGC subcluster and injury condition (FDR ≤ 0.05, fold-change ≥ 2). (**F**) t-SNE plots of pooled SGC colored by unbiased clustering, separated by injury condition. (**G**) t-SNE plots overlay for *Gfap* expression (blue), separated by injury condition. (**H**) Representative images

*Figure 6 continued on next page*

*Figure 6 continued*

of immunofluorescence staining of DRG sections labeled for GFAP (green) and TUJ1 (magenta) from naïve, SNC, DRC, and SCI conditions. n = 3 biologically independent animals. Scale bar: 100 µm (**I**) Quantification of the percentage of neurons with GFAP (green) positive SGC around them out of all TUJ1-positive neurons (magenta). n = 3 biologically independent animals. One-way analysis of variance (ANOVA) followed by Bonferroni's multiple comparisons test. Data are presented as mean values ± SEM (**J**) Enriched signaling pathways (GO Molecular Function and KEGG 2019) for top differentially expressed genes in subcluster 6. (**K**) Enriched TF (ENCODE and ChEA) in top differentially expressed genes in subcluster 6. (**L**) Protein-protein interaction of top TF expressed in subcluster 6 (STRING).

The online version of this article includes the following figure supplement(s) for figure 6:

**Source data 1.** Source files for scRNAseq analysis; SGC subtypes injury marker genes, DEG in SGC subclusters in all injuries (FDR ≤ 0.05, fold-change ≥2), cluster distribution.

cluster-specific genes (*Figure 6D*). Analysis of DE genes for every subcluster in each injury condition revealed that the majority of gene expression changes occurred following SNC in all subclusters, with the highest changes in subcluster 4 (FDR ≤ 0.05, fold-change ≥2) (*Figure 6E*, *Figure 6—source data 1*). GFAP is a known marker of injured SGC (*Christie et al., 2015*; *Woodham et al., 1989*; *Xie et al., 2009*) and *GFAP* expression was observed in cluster 6 after SNC, but also after DRC in clusters 2, 3, and 4 (*Figure 6F and G*). Immunostaining in DRG sections confirmed that ~ 25 % of neurons in both SNC and DRC conditions were surrounded by GFAP expressing SGC, with no changes after SCI (*Figure 6H,I* and *Figure 6—source data 1*). These results suggest that *GFAP* expression is a marker for SGC injury but does not entirely relate to the different axon regenerative capabilities in peripheral nerve and dorsal root. We next performed GO and KEGG pathway analysis of cluster six marker genes, which revealed enrichment for pathways involved in axon regeneration, calcium signaling pathway, and mineral absorption (*Figure 6J* and *Figure 6—source data 1*). To further characterize the unique cluster 6 marker genes induced by SNC, we performed a transcription factor binding site analysis, which revealed enrichment for *Rest*, *Trp53*, *Rad21*, *Ctcf,* and *Zeb1* (*Figure 6K*). We next used STRING to determine the functional protein interaction of these transcription factors and found that the transcription repressor CTCF was highly associated with RAD21 and p53, *less with* ZEB1 and not at all with REST (*Figure 6L*). *Zeb1* is known to control epithelial to mesenchymal transition (EMT) leading to a more plastic state (*Zhang et al., 2015*), while *Rest* is involved in the signaling pathways regulating pluripotency (*Singh et al., 2008*), suggesting that cluster 6 adopts a more plastic state after SNC that might play a role in nerve regeneration.

## Activation of PPARα with fenofibrate increases axon regeneration after dorsal root crush

We recently revealed that PPARα signaling downstream of FASN in SGC promotes axon regeneration after peripheral neve injury (*Avraham et al., 2020*). To determine the overall biological differences in SGC responses to peripheral and central injuries we examined the up and downregulated biological processes and signaling pathways enriched in pooled SGC in each injury condition (*Figure 7A*, *Figure 7—figure supplement 1A* and *Figure 7—source data 1*). Following nerve crush, SGC upregulate processes involving macrophage chemotaxis and migration, with upregulation of the genes *Ccl5*, *Dock8*, *Cmklr1* and *Lbp*, that might assist in the macrophage expansion in the DRG (*Figure 7—figure supplement 1A,B* and *Figure 7—source data 1*). In contrast, after DRC, SGC upregulate genes involve in negative regulation of axon extension and guidance and negative regulation of chemotaxis (*Figure 7—figure supplement 1A* and *Figure 7—source data 1*), which might relate, in part, to the slow axonal regeneration following DRC. After SCI, SGC upregulate genes involved in ECM assembly, myelination and chemical synaptic transmission (*Figure 7—figure supplement 1A*). In agreement with our recent studies (*Avraham et al., 2020*), KEGG pathway analysis indicate that SGC upregulate fatty acid biosynthesis and PPARα signaling pathway after SNC, with upregulation of the PPARα target genes *Hmgcs2* and *Scd1* (*Figure 7A and B*). However, none of these pathways were enriched after DRC and PPARα signaling was downregulated after SCI (*Figure 7A*). Plotting all cells expressing PPARα revealed enriched expression in the SGC cluster across all injury conditions, with the highest distribution in SGC subclusters 2 , 3 and 6 (*Figure 7C and D* and *Figure 7—source data 1*). Additionally, PPARα target genes were also enriched in the SGC cluster (*Avraham et al., 2020* and *Figure 7—figure supplement 1C*).

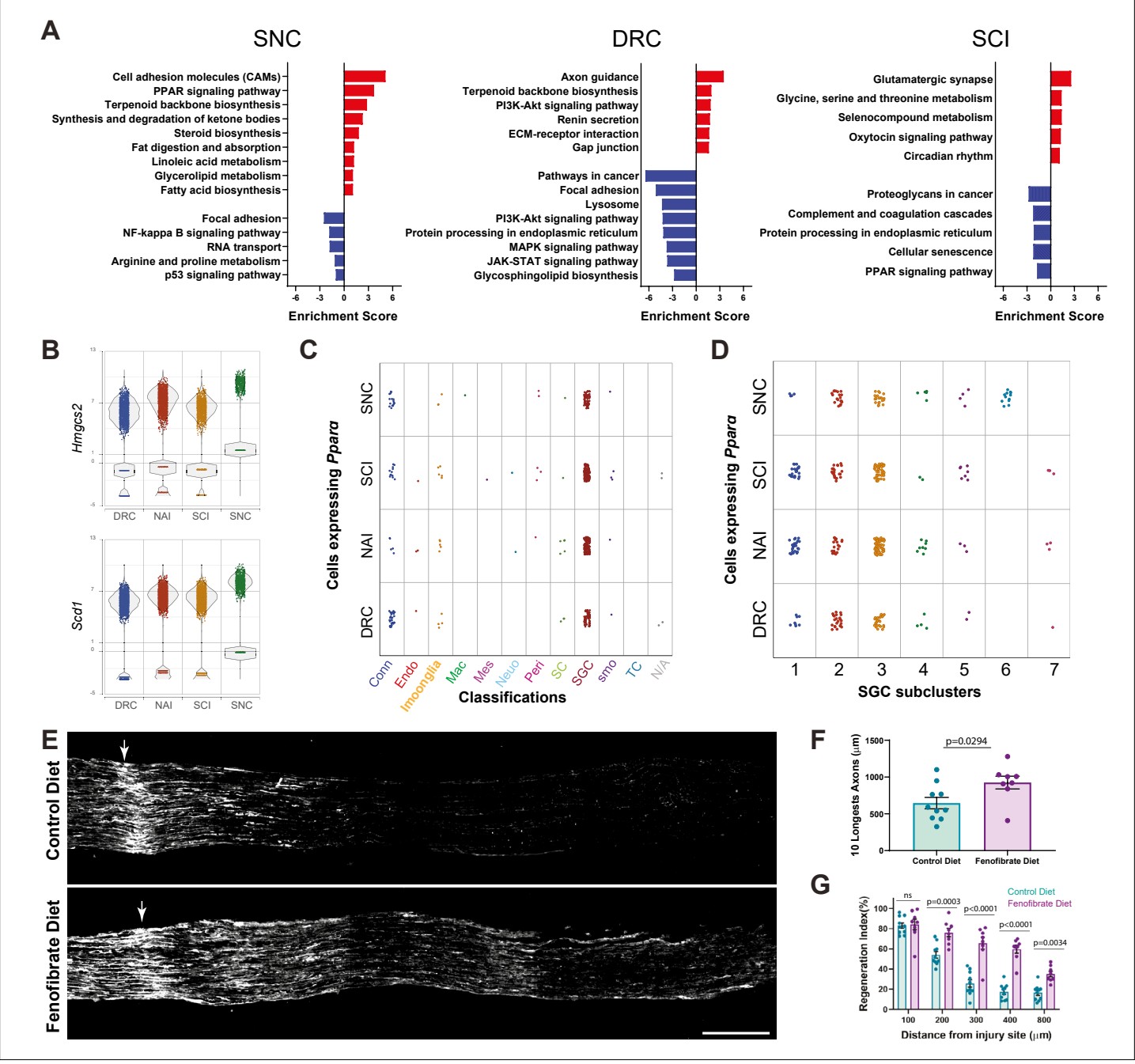

**Figure 7.** Activation of PPARα with fenofibrate increases axon regeneration after dorsal root crush. (**A**) Enriched signaling pathways (KEGG 2019) of differentially upregulated (red) and downregulated (blue) genes in the SGC cluster. n = 2 biologically independent experiments. (FDR ≤ 0.05, fold-change ≥ 2). (**B**) Violin plots for expression of PPARα target genes *Hmgcs2* and *Scd1* across all injuries. (**C**) Plot of *Pparα* expressing cells across all injury conditions. (**D**) Plot of *Pparα* expressing cells in SGC subclusters across all injury conditions. (**E**) Representative longitudinal sections of dorsal roots 3 days after injury from mice fed with fenofibrate or control diet, stained for SCG10. Arrows indicate the crush site, Scale Bar: 100 μm. (**F**) Length of the longest 10 axons was measured in 10 sections for each nerve. Unpaired t-test. n = 10 (control diet) and n = 8 (Fenofibrate diet) biologically independent animals. Data are presented as mean values ± SD (**G**) Regeneration index was measured as SGC10 intensity normalized to the crush site. Two-way ANOVA followed by Bonferroni's multiple comparisons test. n = 10 (control diet) and n = 8 (Fenofibrate diet) biologically independent animals. Data are presented as mean values ± SEM.

The online version of this article includes the following figure supplement(s) for figure 7:

**Source data 1.** Source files for scRNAseq analysis; DEG in SGC (FDR ≤ 0.05, fold-change ≥ 2), PPARαexpression in all cell types.

**Figure supplement 1.** Distinct response of SGC to peripheral vs. central injuries.

We previously showed that the specific PPARα agonist fenofibrate, an FDA-approved drug used to treat dyslipidemia with minimal activity toward PPARγ (**Kim et al., 2016**; **Rosenson, 2008**), upregulated PPARα target genes in the DRG and rescued the impaired axon growth in mice lacking fatty acid synthase in SGC (**Avraham et al., 2020**). Since SGC do not activate PPARα after DRC (**Figure 7A**), a model in which axonal growth occurs at about half the rate of peripheral axons (**Oblinger and Lasek, 1984**; **Wujek and Lasek, 1983**), we tested if fenofibrate treatment improved axon regeneration after DRC. Mice were fed with fenofibrate or control diet for 2 weeks as described (**Avraham et al., 2020**) and then underwent DRC injury. We measured the extent of axon regeneration past the injury site three days later by labeling dorsal root sections with SCG10, a marker for regenerating axons (**Shin et al., 2014**; **Figure 7E**). The crush site was determined according to highest SCG10 intensity along the nerve. First, we measured the length of the 10 longest axons, which reflect the extent of axon elongation, regardless of the number of axons that regenerate (**Figure 7F**). Second, we measured a regeneration index by normalizing the average SCG10 intensity at distances away from the crush site to the SCG10 intensity at the crush site (**Figure 7G**). This measure takes into accounts both the length and the number of regenerating axons past the crush site. Both measurement methods revealed improved regeneration in mice treated with fenofibrate compared to control diet (**Figure 7F and G** and **Figure 7—source data 1**). These results indicate that the lack of PPARα signaling in SGC after central axon injury contributes to the decreased regenerative ability. Whether PPARα activation could also promote axon entry passed the dorsal root entry zone and improve axon regeneration of dorsal column axons after SCI remains to be determined. This study provides a map of the distinct DRG microenvironment responses to peripheral and central injuries at the single-cell level and highlights that manipulating SGC in the DRG could lead to avenues to promote functional recovery after CNS injuries.

## Discussion

Our unbiased single-cell approach fills a critical gap in knowledge for the field and enables in depth characterization of the molecular profile of cells comprising the neuronal microenvironment in the DRG following acute peripheral and central injuries. Our analysis demonstrates major, yet distinct molecular changes in non-neuronal cells in response to peripheral nerve injury and dorsal root injury, with more limited responses after SCI. It is possible that the percent of DRG neurons lesioned under any injury paradigms may impact the injury response of the microenvironment. Nonetheless, all three injury paradigms are widely used models to study the mechanisms promoting axon regeneration. Our study highlights that manipulating non-neuronal cells could lead to avenues to promote functional recovery after CNS injuries or disease.

In endothelial cells, pericytes, Schwann cells, macrophages and SGC, gene expression changes were the largest after peripheral injury, but also occurred after dorsal root crush and spinal cord injury. How non-neuronal cells in the DRG sense a distant axon injury remains poorly understood. The mechanisms underlying SGC responses to nerve injury were proposed to depend on early spontaneous activity in injured neurons as well as retrograde signaling and direct bidirectional communication (**Christie et al., 2015**; **Xie et al., 2009**). DLK was shown to be important for retrograde injury signaling in sensory neurons (**Shin et al., 2019**), but whether DLK also regulate SGC response to injury remains to be determined. Macrophages have been proposed to respond to chemokines expressed by injured neurons such as CCL2 for their recruitment to the DRG after nerve injury (**Niemi et al., 2016**; **Niemi et al., 2013**) and DLK was shown to regulate the expression of cytokines such as CCL2 in neurons (**Hu et al., 2019**). Injured sensory neurons also express colony-stimulating factor 1 (CSF1) in a DLK-dependent manner (**Guan et al., 2016**; **Wlaschin et al., 2018**), which could contribute to recruit CSF1R expressing macrophages to the DRG. Whether DLK activity in neurons regulate communication to immune cells directly or via SGC will require further investigations. Our observation that SGC express CCL5 after nerve injury suggests that SGC may contribute to recruit macrophages to the DRG.

The breakdown of the BNB in response to nerve damage can lead to neuronal dysfunction and contribute to the development of neuropathy (**Richner et al., 2018**). While the impact of physical nerve damage or disease state such as diabetic neuropathy on the BNB are being studied (**Richner et al., 2018**), whether and how nerve injury affect the BNB in the sensory ganglion is not known. BNB leakage can give blood derived molecules direct access to sensory neurons and promote infiltration of

inflammatory cells to engage inflammatory responses. Our results suggest that endothelial cells and pericytes respond differently to peripheral and central injuries, with alteration in expression of tight junction related genes, potentially underlying changes in BNB permeability. Similar to the situation in the nerve, Schwann cells and potentially macrophages may secrete factors such as VEGF and cytokines, altering endothelial cell function. Sympathetic innervation of blood vessels in the DRG following nerve injury, which depends in part on IL-6 signaling, may also underlie the changes observed in endothelial cells (*Ramer and Bisby, 1998a*; *Ramer et al., 1998c*).

In the injured nerve, Schwann cells guide regenerating axons to support distal innervation (*Gomez-Sanchez et al., 2017*; *Jessen and Mirsky, 2016*). To accomplish these regenerative functions, Schwann cells are reprogrammed to a repair state, which relies in part on the transition from an epithelial fate to a more plastic mesenchymal fate (*Arthur-Farraj et al., 2017*; *Clements et al., 2017*). Our studies highlight that away from the injury site in the DRG, Schwann cells respond to injury in part by regulating the hippo pathway, a central pathway in cellular growth and plasticity (*Yu and Guan, 2013*; *Zhao et al., 2011*). The transcription factor *Yap*, which is regulated by the hippo pathway, is upregulated after nerve injury but downregulated after both central injuries. The Nf2-Yap signaling was shown to play important roles in controlling the expansion of DRG progenitors and glia during DRG development (*Serinagaoglu et al., 2015*). These results suggest that Schwann cells in the DRG respond differently to distant peripheral and central axon injuries, and that central axon injury may limit their plasticity. Prior studies demonstrated that glial overexpression of NGF enhances neuropathic pain and adrenergic sprouting into DRG following chronic sciatic constriction in mice (*Ramer et al., 1998b*). Since *Ngf* is downregulated in Schwann cell after all injuries, Schwann cells in the DRG may regulate pain and adrenergic sprouting after injury.

Macrophages are known to regulate regenerative responses. In the nerve, macrophages function primarily to assist Schwann cells for debris clearance. Nerve resident macrophages in naive conditions share features with CNS microglia (*Wang et al., 2020*; *Ydens et al., 2020*). In the injured nerve, resident macrophages represent only a small subset that secretes chemoattractants to recruit circulating monocytes-derived macrophages (*Ydens et al., 2020*). These recruited monocytes-derived macrophages express *Arg1* and represent the main macrophage population guiding nerve repair (*Ydens et al., 2020*). In the naive DRG, we found that resident macrophages share similar properties with endoneurial macrophages in the nerve. However, *Arg1* expression is only observed after dorsal root crush and not after peripheral nerve injury or SCI. The increase in cell cycle marker after nerve injury suggest that the increase in macrophage numbers in the DRG results largely from macrophage proliferation. The increase in IL-6, which has been associated with nerve regeneration (*Cafferty et al., 2004*; *Cao et al., 2006*), is associated only with nerve injury and not central injuries. Immune cells in the DRG may thus not follow a strict classification and complex subtypes exist in naive conditions that are differentially regulated by peripheral and central axon injury.

Our data also unravels the existence of a small proportion of cells that share expression of macrophage and glial genes, which we named Imoonglia. These cells increase in number after all injuries. Previous studies suggested that after injury some macrophages penetrate into the space between SGC and neurons (*Hu and McLachlan, 2002*). Several other studies interpreted the co-localization of SGC and immune markers as evidence that a subset of SGC express an immune cell character in human (*van Velzen et al., 2009*), rat (*Donegan et al., 2013*; *Jasmin et al., 2010*), mice (*Huang et al., 2021*), and canine ganglion (*Huang et al., 2021*). Interestingly, rat SGC express CD45, but unlike in humans, the CD45-positive cells were only present after nerve injury (*Jasmin et al., 2010*). Recent scRNAseq studies further support the presence of a populations of SGC enriched in immune-response genes (*Mapps et al., 2021*; *van Weperen et al., 2021*). Whether immune cells move to a peri-neuronal position and become Imoonglia or whether SGCs transition from a progenitor state to acquire an immune signature in stressful condition such as after nerve injury remains to be tested. Our single-cell profiling, flow cytometry, and immunofluorescence data suggest that Imoonglia represent a specific cell type that is increased after injury and share the spatial arrangement of SGC surrounding sensory neurons. This may be similar to astrocytes in the CNS, which in addition to their homeostatic functions can undergo an inflammatory transcriptional transition following inflammation after acute insults like stroke (*Zamanian et al., 2012*), spinal cord injury (*Karimi-Abdolrezaee and Billakanti, 2012*), and systemic inflammation (*Hasel et al., 2021*). It is tempting to speculate that Imoonglia may function in immune surveillance in the DRG and specifically in the case of viral infection. Indeed, SGC

were suggested to restrict the local diffusion of viruses such as herpes simplex virus, varicella zoster virus, HIV-1 and haemagglutinating encephalomyelitis virus, which belongs to the coronavirus family (*Hanani and Spray, 2020*).

SGC have been previously characterized as a neural crest derived uniform glial population that plays a major role in pain (*Hanani and Spray, 2020*). Our single cell analysis reveals that SGC do not represent a uniform population but that several subtypes exist. In addition to the Imoonglia discussed above, we identified four SGC clusters in naive conditions and up to seven clusters after injury. The enrichment of different biological pathways in each cluster suggest that they each have a specialized function. Cluster 3 shares the most similarities with astrocytes and is enriched for PPARα signaling and fatty acid biosynthesis. Cluster 1 and 2 represent the most unique SGC subtypes. Interestingly, cluster 1 is enriched for calcium signaling pathway, which in SGC is known to play a role in pain (*Hanani and Spray, 2020*). It will be important to determine in future studies whether the most unique SGC subtypes (cluster 1 and 2) surround specific neuronal subtypes, or if a mosaic organization exists, with different SGC surrounding the same neuron. The appearance of a specific cluster (cluster 6) after nerve injury suggest that this cluster plays an essential role in nerve injury responses. Cluster six is also enriched for the transcription factor REST, which in astrocyte regulates gliosecretion (*Prada et al., 2011*). Transcription factor binding site analysis of genes in cluster 6 revealed enrichment for the EMT gene *Zeb1*. EMT is often linked to increased plasticity and stem cell activation during tissue regeneration, suggesting that cluster 6 is related to plasticity of SGC.

In neurons, axonal PPARγ contributes to the pro-regenerative response after axon injury (*Lezana et al., 2016*) and our recent study suggest that FASN may generate ligands for PPARγ in neurons (*Ewan et al., 2021*). We recently demonstrated that in SGC, FASN is required for the activation of PPARα and that PPARα signaling promote axon regeneration in adult peripheral nerves (*Avraham et al., 2020*). Here, we showed that this PPARα signaling pathway is not activated in SGC after injury to centrally projecting axons (SCI and DRC). Further, we demonstrate that the FDA approved PPARα agonist fenofibrate increased axon regeneration in the dorsal root, a model of poor sensory axon regeneration. In our previous study, we showed that removing the enzyme FASN, which is upstream of PPARα activation, specifically in SGC, decreases axon growth in the sciatic nerve (*Avraham et al., 2020*). Altogether, these findings suggest that the lack of PPARα activation after DRC contributes to the low regeneration rates of axons in the dorsal root and provide insights into the translational potential of fenofibrate. Indeed, fenofibrate is used clinically to treat lipid disorders, and has unexpectedly been shown in clinical trials to have neuroprotective effects in diabetic retinopathy (*Bogdanov et al., 2015*; *Moreno and Cerù, 2015*) and in traumatic brain injury (*Chen et al., 2007*). In mice, fenofibrate was shown to modestly increase tissue sparing following spinal contusion injury (*Almad et al., 2011*). The neuroprotective role of fenofibrate was also recently observed in a mouse model of paclitaxel chemotherapy-induced peripheral neuropathy (*Caillaud et al., 2021*). The enrichment of biological pathways related to PPARα signaling are largely conserved between rodent and human SGC (*Avraham et al., 2021*). Together, these findings support the notion that PPARα activation is a promising therapeutics for neurologic disease and CNS injury (*Mandrekar-Colucci et al., 2013*). The transcriptional profiling of SGC in response to peripheral and central axon injury highlights that manipulating non-neuronal cells could lead to avenues to treat CNS injuries.

# Materials and methods

**Key resources table**

| Reagent type (species) or resource | Designation | Source or reference | Identifiers | Additional information |
|---|---|---|---|---|
| Strain, strain background (mice, C57Bl/6) | *Sun1*-sfGFP-myc | *Mo et al., 2015* | R26-CAG-LSL-*Sun1*-sfGFP-myc | from Dr. Harrison Gabel |
| Strain, strain background (*M. musculus*, C57Bl/6) | *Actl6b*^Cre | *Zhan et al., 2015* | | from Dr. Harrison Gabel |
| Strain, strain background (*M. musculus*, C57Bl/6) | *Fabp7*^creER | *Maruoka et al., 2011* | | from Dr. Toshihiko Hosoya |

*Continued on next page*

*Continued*

| Reagent type (species) or resource | Designation | Source or reference | Identifiers | Additional information |
|---|---|---|---|---|
| Strain, strain background (*M. musculus*, C57Bl/6) | Aldh1l1::Rpl10a-Egfp | *Doyle et al., 2008* | B6;FVB-Tg(Aldh1l1-EGFP/Rpl10a) JD130Htz/J | from Dr. Joseph Dougherty. |
| Antibody | Rabbit polyclonal anti Glial Fibrillary Acidic Protein | Agilent | Cat# Z033429-2 | IF (1:500) |
| Antibody | Rabbit polyclonal anti Fatty acid binding protein 7 | Thermo Fisher Scientific | Cat# PA5-24949, RRID:AB_2542449 | IF (1:1000) |
| Antibody | Rabbit polyclonal anti STMN2/ SCG10 | Novus | Cat# NBP1-49461, RRID:AB_10011569 | IF (1:1000) |
| Antibody | Mouse monoclonal anti Tubulin beta-3 chain | BioLegend | Cat# 802001, RRID:AB_291637 | IF (1:1000) |
| Antibody | Rabbit polyclonal anti MKI67 | Abcam | cat# ab15580 | IF (1:500) |
| Antibody | Mouse monoclonal anti CD68 | Bio-Rad | Cat# MCA1957 clone;FA-11 | IF (1:1000) |
| Antibody | Rat monoclonal anti EMR1/ F4/80-PE | eBioscience | Cat# 5010786 | FC (1:1000) |
| Antibody | Mouse Recombinant anti CD16/CD32 | Biolegend | Cat# 158,302 Clone QA17A34 | FC (1:50) |
| Antibody | Rat monoclonal anti EMR1/ F4/80-BV605 | Biolegend | Cat# 123,133 Clone BM8 | FC (1:200) |
| Antibody | Rat monoclonal anti CD11b/ ITGAM-PerCP-Cy5.5 | Biolegend | Cat# 101,228 Clone M1/70 | FC (1:200) |
| Antibody | Rat monoclonal anti CD45-BV750 | BD Biosciences | Cat# 746,947 Clone 30-F11 | FC (1:200) |
| Sequence-based reagent | *Rpl13a* Forward | PrimerBank | PCR primer ID 334688867c2 | AGCCTACCAGAAAGTTTGCTTAC |
| Sequence-based reagent | *Rpl13a* Reverse | PrimerBank | PCR primer ID 334688867c2 | GCTTCTTCTTCCGATAGTGCATC |
| Sequence-based reagent | *Cd74* Forward | PrimerBank | PCR primer ID 13097486a1 | AGTGCGACGAGAACGGTAAC |
| Sequence-based reagent | *Cd74* Reverse | PrimerBank | PCR primer ID 13097486a1 | CGTTGGGGAACACACACCA |
| Sequence-based reagent | *H2-Aa* Forward | PrimerBank | PCR primer ID 31981716a1 | TCAGTCGCAGACGGTGTTTAT |
| Sequence-based reagent | *H2-Aa* Reverse | PrimerBank | PCR primer ID 31981716a1 | GGGGGCTGGAATCTCAGGT |
| Sequence-based reagent | *Ctss* Forward | PrimerBank | PCR primer ID 10946582a1 | CCATTGGGATCTCTGGAAGAAAA |
| Sequence-based reagent | *Ctss* Reverse | PrimerBank | PCR primer ID 10946582a1 | TCATGCCCACTTGGTAGGTAT |
| Sequence-based reagent | *Ccl2* Forward | PrimerBank | PCR primer ID 6755430a1 | TTAAAAACCTGGATCGGAACCAA |
| Sequence-based reagent | *Ccl2* Reverse | PrimerBank | PCR primer ID 6755430a1 | GCATTAGCTTCAGATTTACGGGT |
| Sequence-based reagent | *Il1b* Forward | PrimerBank | PCR primer ID 6680415a1 | GCAACTGTTCCTGAACTCAACT |
| Sequence-based reagent | *Il1b* Reverse | PrimerBank | PCR primer ID 6680415a1 | ATCTTTTGGGGTCCGTCAACT |
| Sequence-based reagent | *Tnf* Forward | PrimerBank | PCR primer ID 7305585a1 | CCCTCACACTCAGATCATCTTCT |
| Sequence-based reagent | *Tnf* Reverse | PrimerBank | PCR primer ID 7305585a1 | GCTACGACGTGGGCTACAG |
| Sequence-based reagent | *Mki67* Forward | PrimerBank | PCR primer ID 1177528a1 | ATCATTGACCGCTCCTTTAGGT |

*Continued on next page*

*Continued*

| Reagent type (species) or resource | Designation | Source or reference | Identifiers | Additional information |
|---|---|---|---|---|
| Sequence-based reagent | *Mki67* Reverse | PrimerBank | PCR primer ID 1177528a1 | GCTCGCCTTGATGGTTCCT |
| Commercial assay or kit | High Capacity cDNA Reverse Transcription kit | Applied Biosystems | Cat# 4368814 | |
| Commercial assay or kit | Gel Bead and Library Kit | 10 x Genomics | GemCode Single-Cell 3' | |
| Chemical compound, drug | Lycopersicon esculentum (tomato) lectin | Vector lab | Cat# DL-1178–1 | 100 ul |
| Chemical compound, drug | Trizol | Thermo Fisher | Cat #15596026 | |
| Chemical compound, drug | Tamoxifen | Envigo Teklad | TD.130858 | Chow pellet 500 mg per kg |
| Chemical compound, drug | Fenofibrate | Envigo Teklad | Sigma Cat# F6020 | Chow pellet 0.2% |
| Chemical compound, drug | PowerUp SYBR Green master mix | Thermo Fisher | Cat #a25742 | |
| Software, algorithm | Partek Flow | Partek | Build version 9.0.20.0417 | |
| Software, algorithm | Nikon-NIS Elements | Nikon | Version 4.60 | |
| Software, algorithm | Prism | GraphPad | Prism8 | |
| Software, algorithm | Fiji | ImageJ | | |
| Software, algorithm | FlowJo | Tree Star | | |

## Animals and procedures

All animals were approved by the Washington University School of Medicine Institutional Animal Care and Use Committee (IACUC) under protocol A3381-01. All experiments were performed in accordance with the relevant guidelines and regulations. All experimental protocols involving mice were approved by Washington University School of Medicine (protocol #20180128). Mice were housed and cared for in the Washington University School of Medicine animal care facility. This facility is accredited by the Association for Assessment & Accreditation of Laboratory Animal Care (AALAC) and conforms to the PHS guidelines for Animal Care. Accreditation - 7/18/97, USDA Accreditation: Registration # 43 R-008.

During surgery, 8- to 12 -week-old female C57Bl/6 mice were anesthetized using 2 % inhaled isoflurane. Sciatic nerve injuries were performed as previously described (*Avraham et al., 2020*; *Cho et al., 2015*; *Cho et al., 2013*). Briefly, the sciatic nerve was exposed with a small skin incision (~1 cm) and crushed for 5 s using #55 forceps. The wound was closed using wound clips and injured L4 and L5 dorsal root ganglia were dissected at the indicated time post-surgery. Contralateral DRG served as uninjured control. For spinal cord injury (SCI), a small midline skin incision (~1 cm) was made over the thoracic vertebrae at T9–T10, the paraspinal muscles freed, and the vertebral column stabilized with metal clamps placed under the T9/10 transverse processes. Dorsal laminectomy at T9/10 was performed with laminectomy forceps, the dura removed with fine forceps, and the dorsal column transversely cut using fine iridectomy scissors. Dorsal root injury was performed similarly as SCI, except that procedures were performed at the L2-L3 vertebral level, and fine forceps used to crush the right L4 and L5 dorsal roots for 5 s.

L4 and L5 roots are in close proximity anatomically hence both roots were crushed simultaneously where the distance from the crush site to L4 DRG is 4–5 mm and 7–8 mm to L5 DRG. During dorsal root crush, the roots are forcefully squeezed causing the disruption of nerve fibers without interrupting the endoneurial tube.

## Mouse strains

Eight- to 12 -week-old male and female mice were used for all experiments, except for scRNAseq experiment, where only C57Bl/6 females were used. The *Sun1-sfGFP-myc* (INTACT mice: *R26-CAG-LSL-Sun1-sfGFP-myc*) (*Mo et al., 2015*), and *Actl6b*^Cre (*Baf53b*) (*Zhan et al., 2015*) was a generous gift from Dr. Harrison Gabel. The Aldh1l1::Rpl10a-Egfp transgenic line (B6;FVB-Tg(Aldh1l1-EGFP/Rpl10a)

JD130Htz/J) (*Doyle et al., 2008*) was a generous gift from Dr. Joseph Dougherty. The *Fabp7*[creER] transgenic line (*Maruoka et al., 2011*) was a generous gift from Dr. Toshihiko Hosoya.

## Single-cell RNAseq

L4 and L5 DRG's from mice 8–12 weeks old were collected into cold Hank's balanced salt solution (HBSS) with 5 % Hepes, then transferred to warm Papain solution and incubated for 20 min in 37 °C. DRG's were washed in HBSS and incubated with Collagenase for 20 min in 37 °C. Ganglia were then mechanically dissociated to a single-cell suspension by triturating in culture medium (Neurobasal medium), with Glutamax, PenStrep and B-27. Cells were washed in HBSS+ Hepes + 0.1% BSA solution, passed through a 70 micron cell strainer. Hoechst dye was added to distinguish live cells from debris and cells were FACS sorted using MoFlo HTS with Cyclone (Beckman Coulter, Indianapolis, IN). Sorted cells were washed in HBSS+ Hepes + 0.1% BSA solution and manually counted using a hemocytometer. Solution was adjusted to a concentration of 500 cell/microliter and loaded on the 10 X Chromium system. Single-cell RNA-Seq libraries were prepared using GemCode Single-Cell 3′ Gel Bead and Library Kit (10 x Genomics). A digital expression matrix was obtained using 10 X's CellRanger pipeline (Washington University Genome Technology Access Center). Quantification and statistical analysis were done with the Partek Flow package (Build version 9.0.20.0417).

Filtering criteria: Low-quality cells and potential doublets were filtered out from analysis using the following parameters; total reads per cell: 600–15,000, expressed genes per cell: 500–4000, mitochondrial reads <10%. A noise reduction was applied to remove low expressing genes ≤ 1 count. Counts were normalized and presented in logarithmic scale in CPM (count per million) approach. An unbiased clustering (graph-based clustering) was done and presented as t-SNE (t-distributed stochastic neighbor embedding) plot, using a dimensional reduction algorithm that shows groups of similar cells as clusters on a scatter plot. Differential gene expression analysis performed using an ANOVA model; a gene is considered differentially-expressed (DE) if it has an FDR ≤ 0.05 and a fold-change ≥2. The data was subsequently analyzed for enrichment of GO terms and the KEGG pathways using Partek flow pathway analysis.

A differential trajectory map of single cells was performed using 'Monocle2' with standard setting. The algorithm orders a set of individual cells along a path / trajectory / lineage and assign a pseudo-time value to each cell that represents where the cell is along that path. This method facilitates the discovery of genes that identify certain subtypes of cells, or that mark intermediate states during a biological process as well as bifurcate between two alternative cellular fates. Partek was also used to generate figures for t-SNE, scatter plot and trajectory analysis representing gene expression.

Cell-cell interaction analysis was performed based on CellPhoneDB repository (v2.1.6),which was developed for human, the mouse genes were converted to human genes first. Statistical iterations were set at 1,000 and gene expressed by less than 10 % of cells in the cluster were removed. Network visualization was performed with Cytoscape (v3.8.2) using the identified significant interactions between the clusters.

## Flow cytometry

Ganglia were enzymatically and mechanically dissociated to a single-cell suspension as described above. For neuronal detection, hoechst dye was added to distinguish live cells from debris. Cells were analyzed on a Attune NxT flow cytometer (ThermoFisher Scientific). For Imoonglia detection, single cell suspensions were incubated for 15 min at 4 °C in Zombie NIR Fixable viability dye (Biolegend) diluted 1:500 in PBS, centrifuged at 420 x g for 5 min and resuspended in anti-CD16/CD32 (Fc Block; Biolegend) diluted 1:50 in FACS buffer (PBS with 2 % BSA and 1 mM EDTA) for five minutes at 4 °C to block Fc receptors. Cells were then incubated with F4/80-BV605, CD11b-PerCP-Cy5.5, and CD45-BV750 antibodies (all 1:200 dilution) for 10 min at room temperature in FACS buffer, centrifuged at 420 x g for 5 min and resuspended in FACS buffer. Samples were run on a Cytek Aurora spectral cytometer (Cytek). Data was analyzed using FlowJo software (Tree Star).

## TEM

Mice were perfused with 2.5 % glutaraldehyde with 4 % paraformaldehyde in 0.1 M Cacodylate buffer, followed by post fix. A secondary fix was done with 1 % osmium tetroxide. For Transmission electron microscopy (TEM), tissue was dehydrated with ethanol and embedded with spurr's resin. Thin sections

(70 nm) were mounted on mesh grids and stained with 8 % uranyl acetate followed by Sato's lead stain. Sections were imaged on a Jeol (JEM-1400) electron microscope and acquired with an AMT V601 digital camera. (Washington University Center for Cellular Imaging).

## RNA isolation and quantitative PCR

DRG and nerves were lysed and total RNA was extracted using Trizol reagent (Thermo Fisher, Cat #15596026). Next, RNA concentration was determined using a NanoDrop 2000 (Thermo Fisher Scientific). First strand synthesis was then performed using the High Capacity cDNA Reverse Transcription kit (Applied Biosystems). Quantitative PCR was performed using PowerUp SYBR Green master mix (Thermo Fisher, Cat #a25742) using 5 ng of cDNA per reaction. Plates were run on a QuantStudio 6 Flex and analyzed in Microsoft Excel. The average Ct value from three technical replicates was averaged normalized to the internal control *Rpl13a*. All primer sequences were obtained from PrimerBank (Harvard) and product size validated using agarose gel electrophoresis.

## Tissue preparation and immunohistochemistry

After isolation of either dorsal root or DRG, tissue was fixed using 4 % paraformaldehyde for 1 hr at room temperature. Tissue was then washed in PBS and cryoprotected using 30 % sucrose solution at 4 °C overnight. Next, the tissue was embedded in O.C.T., frozen, and mounted for cryosectioning. All frozen sections were cut to a width of 10 μm for subsequent staining. For immunostaining of DRG and nerve sections, slides were washed 3 x in PBS and then blocked in solution containing 10 % goat serum in 0.2 % Triton-PBS for 1 hr. Next, sections were incubated overnight in blocking solution containing primary antibody. The next day, sections were washed 3 x with PBS and then incubated in a blocking solution containing a secondary antibody for 1 hr at room temperature. Finally, sections were washed 3 x with PBS and mounted using ProLong Gold antifade (Thermo Fisher Scientific). Images were acquired at 10 x or 20 x using a Nikon TE2000E inverted microscope and images were analyzed using Nikon Elements. Antibodies were as follow: SCG10/Stmn2 (1:1000; Novus catalog #NBP1-49461, RRID:AB_10011569), Tubb3/βIII tubulin antibody (BioLegend catalog #802001, RRID:AB_291637), Fabp7 (Thermo Fisher Scientific Cat# PA5-24949, RRID:AB_2542449), MKI67 (Abcam cat# ab15580), CD68 (Bio-Rad Cat# MCA1957 clone;FA-11), GFAP (Agilent, Cat# Z033429-2). Stained sections with only secondary antibodies were used as controls. For Lectin injection, mice were deeply anesthetized by 1.5 % isoflurane. A total of 100 μl of Lycopersicon esculentum (tomato) lectin (vector lab; Catalog#DL-1178–1) was used per mouse by injection into the tail vein. Mice were sacrificed after 20 min of injection. Samples were collected following the procedure described above. Twenty μm thickness of DRG cryosections were used for immunofluorescence staining and image was captured under LSM880 confocal microscope.

## Data collection and analyses

Data collection and analyses were performed blind to the conditions of the experiments. Single-cell RNAseq analysis was performed in an unbiased manner using established algorithms.

## Quantification and statistical analysis

Quantifications were performed by a blinded experimenter to injury type and treatment. Fiji (ImageJ) analysis software was used for immunohistochemistry images quantifications. Nikon Elements analysis software was used to trace regenerating axons in the dorsal root sections. Statistics was performed using GraphPad (Prism8) for t-test and one/two-way ANOVA followed by Bonferroni's multiple comparisons test. Error bars indicate the standard error of the mean (SEM). Heatmaps were calculated as fold change of normalized counts compared to naive or as z-scores. The formula for calculating z-score used $z = (x-\mu)/\sigma$, where x is the expression of the gene in all cells for each condition, μ is the sample mean of the gene in all conditions, and σ is the sample standard deviation in all conditions. Sample was calculated as the average expression of cells in each condition.

## Acknowledgements

We thank members of the Cavalli lab for valuable discussions. We thank Mayssa Mokalled for helpful discussion and comments on glial cells and spinal cord injury. We thank Jonathan Kipnis for constructive discussions and precious help with flow cytometry. We thank Harrison Gabel for his generous gift

of the *Baf53b*-cre mouse line and Joseph Dougherty for providing the Aldh1l1::Rpl10a-Egfp transgenic line. We gratefully acknowledge Greg Strout, Ross Kossina and Dr. James Fitzpatrick from the Washington University Center for Cellular Imaging (WUCCI), which is supported in part by Washington University School of Medicine, The Children's Discovery Institute of Washington University, and St. Louis Children's Hospital (CDI-CORE-2015–505 and CDI-CORE-2019–813) and the Foundation for Barnes-Jewish Hospital (3770) for assistance in acquiring and interpreting Transmission Electron Microscopy (TEM) data. We thank the Bioinformatics Research Core for CellPhone DB analysis. We also thank Anushree Seth and Madison Mack in association with InPrint for illustration in *Figure 1d*. This work was funded in part by a post-doctoral fellowship from The McDonnell Center for Cellular and Molecular Neurobiology to OA, by The McDonnell Center for Cellular and Molecular Neurobiology, NIH grant R35 NS122260, R01 NS111719 and R21 NS115492 to V.C.

## Additional information

### Funding

| Funder | Grant reference number | Author |
|---|---|---|
| National Institute of Neurological Disorders and Stroke | NS111719 | Valeria Cavalli |
| National Institute of Neurological Disorders and Stroke | NS115492 | Valeria Cavalli |
| National Institute of Neurological Disorders and Stroke | NS122260 | Valeria Cavalli |
| The McDonnell Center for Cellular and Molecular Neurobiology | | Valeria Cavalli Oshri Avraham |

The funders had no role in study design, data collection and interpretation, or the decision to submit the work for publication.

### Author contributions
Oshri Avraham, Conceptualization, Data curation, Formal analysis, Methodology, Validation, Writing – original draft, Writing – review and editing, Investigation; Rui Feng, Formal analysis, Methodology, Software, Validation, Visualization, Conceptualization, Investigation; Eric Edward Ewan, Conceptualization, Investigation, Methodology; Justin Rustenhoven, Conceptualization, Formal analysis, Methodology; Guoyan Zhao, Data curation, Methodology, Resources, Software; Valeria Cavalli, Conceptualization, Funding acquisition, Project administration, Supervision, Writing – original draft, Writing – review and editing

### Author ORCIDs
Guoyan Zhao http://orcid.org/0000-0001-5615-6774
Valeria Cavalli http://orcid.org/0000-0001-9978-050X

### Ethics
All animals were approved by the Washington University School of Medicine Institutional Animal Care and Use Committee (IACUC) under protocol A3381-01. All experiments were performed in accordance with the relevant guidelines and regulations. All experimental protocols involving mice were approved by Washington University School of Medicine (protocol #20180128). Mice were housed and cared for in the Washington University School of Medicine animal care facility. This facility is accredited by the Association for Assessment & Accreditation of Laboratory Animal Care (AALAC) and conforms to the PHS guidelines for Animal Care. Accreditation - 7/18/97, USDA Accreditation: Registration # 43-R-008.

Decision letter and Author response
Decision letter https://doi.org/10.7554/eLife.68457.sa1
Author response https://doi.org/10.7554/eLife.68457.sa2

## Additional files

### Supplementary files
• Transparent reporting form

### Data availability
The raw Fastq files and the processed filtered count matrix for scRNA sequencing were deposited at the NCBI GEO database under the accession number GSE158892. Processed data are also available for visualization and download at https://mouse-drg-injury.cells.ucsc.edu/. Data analysis and processing was performed using commercial code from Partek Flow package at https://www.partek.com/partek-flow/.

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
