## [Decision Letter]

**Acceptance summary:**

The authors study differences in dorsal root ganglia cells for their regenerative capacity by comparing a central versus a peripheral nervous system injury, as these cells have processes that are present in both parts of the nervous system. They make extensive use of single cell transcriptomics, and found dramatic differences in the expression state and regenerative potential.

**Decision letter after peer review:**

Thank you for submitting your article "Profiling sensory neuron microenvironment after peripheral and central axon injury reveals key pathways for neural repair" for consideration by *eLife*. Your article has been reviewed by 2 peer reviewers, and the evaluation has been overseen by a Reviewing Editor and Marianne Bronner as the Senior Editor. The reviewers have opted to remain anonymous.

Essential revisions:

1) A more definitive characterization of the immune glia cells, to exclude engulfment as a possibility. This subset of macrophages with glial properties, it could be simply an artifact of macrophages engulfing satellite glial cells.

2) Demonstrate additional advances in the PPARa findings. Given that PPARa was previously implicated in this effect, additional evidence beyond fenofibrate experiment should be provided. Dorsal column sensory axon regeneration after spinal cord injury would be ideal, but could be very challenging. Or, at least characterizing sensory axon regeneration into DREZ after dorsal root crush. If the results are negative, authors should discuss the limitation of manipulating this pathway alone in the CNS.

3) Functional validation to address the role of PPARa signaling in satellite glial cells since the it was done with pharmacological manipulation.

*Reviewer #1 (Recommendations for the authors):*

In this manuscript, Avraham et al. report their results of profiling different cell types in DRG from mice with different types of injury. In general, injuries occurred in PNS (sciatic nerves or dorsal roots) trigger more drastic effects on nearly all DRG cell types, comparing to those applied to CNS (spinal cord), albeit with some exceptions. Among these responding cell populations is a subset of macrophage expressing a satellite glial cell (SGC) marker, and an another population of SGCs, although their lineage and role remain unknown. Furthermore, fatty acid biosynthesis and PPARa signaling pathways are up-regulated after sciatic nerve injury, but down-regulated after dorsal root injury (again these observations are not verified and the underlying mechanisms are elusive). Application of a PPARa agonist is able to elevate axon regeneration after dorsal root injury. In general, this manuscript has a large amount of data from bioinformatics analysis but with limited functional verifications. Thus their biological meaning is less clear.

In Figure 3, the results imply different signaling involvement in DRG macrophages (cell cycle and DNA replication after SNC and steroid biosynthesis and glycolysis/gluconeogenesis pathways after DRC/SCI). Are these results due to differential resident/infiltrated macrophage in DRG after individual injury types?

It is also intriguing to note that all injuries down-regulate genes related to antigen processing and presentation (Figure 3). This seems an interesting observation as macrophages often exhibit pro-inflammatory responses to injury. These results should be verified with independent methods.

In Figure 4, the authors describe a subset of macrophages expressing glial markers whose numbers become increased after injury. Is it possible that these might be the macrophages with engulfed SGCs? To test this, perhaps the authors could compare the abundance of these type-specific RNAs or use other independent methods (with transgenic mice with GFP-labeled SGCs to see if any GFP signals are in these macrophages).

The results in Figure 5 suggest that SGCs represent different cell populations. Again, their biological meaning remains unknown. An obvious possibility is these clusters might reflect their different activation states. It might be useful to apply single cell trajectory analysis to assess their relationship.

Figure 7, are the regeneration results after PPARa agonist comparable to those after sciatic nerve injury? Such information might provide insights as to its translational potential.

*Reviewer #2 (Recommendations for the authors):*

Suggested experiments:

1) Impact on functional validation will be enhanced significantly if regeneration into and beyond DREZ can be assessed with fenofibrate, especially given the previously published results with SNC.

2) Figure 3e and 3g have different markers. Validation will be enhanced if selected markers in Figure 3e is confirmed with immunostaining.

Suggested discussions, clarification or corrections:

1) Using 1 biological replicate for SNC vs. 2 for other conditions seems unbalanced. Ideally use 2 replicates per condition, and at least discuss implications on using 1 replicate for SNC. For instance, what is the variability among the replicates for data presented in Figure 1f,g?

2) Authors indicated that the magnitude of expression changes using number of differentially expressed genes. With Supplemental Table 2, the cut off was based on 0.05 p value instead of FDR corrected (FDR step up). With FDR corrected p-value, the numbers become 614 (SRC vs. NAI), 222(SCI vs. NAI) and 1276 (SNC vs. NAI). Please change it accordingly as the multiple comparison is necessary for sequencing data.

3) Is PPARa pathway enriched in cluster 6 SGCs, which increases after SNC?

4) The discussion on immune glia cells can be more explicit, even though it is understood that this new cell type may not squarely fit with either macrophages or SGCs. Please discuss if this been implicated in the published literature.

5) Please clarify if the data on SGCs overlap with those from Avraham et al., 2020 (i.g. same data set)?

6) "Here we showed that this PPARa signaling pathway is not activated in SGC after central injures." Is DRC a peripheral or central injury?

7) For DRC experiment, was DRC only performed for L4, but both L4 and L5 DRG were collected for scRNA seq?

8) It seems that the least differentially expressed genes are associated with the most abundant cell types sequenced (SGCs and macrophages). Is this due to a higher sample rates, thus more reproducible/reliable data set? And vice versa?

9) "Whether the different regenerative capacities after peripheral or central injury result in part from a lack of response of macrophages, satellite glial cells (SGC) or other non-neuronal cells in the DRG microenvironment remains largely unknown." Can also be an altered response.

10) Even though fenofibrate may have minimal activity towards PPARa, a previous paper on the neuronal role of the γ isoform is relevant and should be discussed: Lezana et al., Dev Neurobiol 2016. Axonal PPARa promotes neuronal regeneration after injury.

11) "Sciatic nerve crush injures approximately half the axons projecting into the peripheral nerves…" Please clarify why this is the case. The percent of DRG neurons lesioned under any injury paradigms (e.g. SNC vs DRC) may impact the injury response of the microenvironment. Also, SCI only injures large diameter DRG neurons, which may explain some of the differences. Would injury level (T9/10) or the distance from DRG neuronal cell bodies also play a role (e.g. DRC is much closer to the cell bodies)?

12) "Dorsal column lesion of the spinal cord damages the ascending axon branches of most large diameter neurons and leaves the descending axon branches in the spinal cord intact." Needs qualifier, as this statement requires certain conditions to be met (e.g., in this case, T9/10 SCI, L4/5 DRG).

13) "…1 biological replicate for SNC (Figure S1d), with an average of 45,000…" Is Figure S1d the right panel here?

14) Figure 3 yellow or orange?

15) In text, reference to phagosomes should be Figure 3d, not 4d.

16) "video 1,2". There is video 2?

17) Please indicate p value when discussing differentially expressed genes, when appropriate.

18) PT53 should be TP53.

---

## [Author Response]

Essential revisions:1) A more definitive characterization of the immune glia cells, to exclude engulfment as a possibility. This subset of macrophages with glial properties, it could be simply an artifact of macrophages engulfing satellite glial cells.

We agree that it is important to exclude the possibility that immune glia cells simply result from phagocytosis of satellite glial cells by macrophages. We have performed additional experiments and additional analyses that strongly support the existence of a subset of macrophages with glial properties. We performed a flow cytometry experiment, which shows that a subset of genetically labeled satellite glial cells (Fabp7-creER: Sun1 GFP) express the specific macrophages markers Cd11b, F4/80 and cd45. This new data is presented in new Figure 4E. We also included additional analyses showing that these immune glial cells express progenitor cell markers (Dhh, *Sox2* and Foxd3) (new figure 4C). We have expanded the methods section to clarify that duplicate cells are filtered out from downstream analysis and also plotted total counts in all cell types (Figure 4—figure supplement 1B), further excluding the possibility that these cells are satellite glial cells engulfed by macrophages. We have also added a trajectory analysis demonstrating that immune glial cells express a transcriptome that position them between satellite glial cells and macrophages (new figure 4D). We believe that these additional experiments and analyses strongly support the characterization of this immune glia cell population, which we now have renamed Imoonglia.

2) Demonstrate additional advances in the PPARa findings. Given that PPARa was previously implicated in this effect, additional evidence beyond fenofibrate experiment should be provided. Dorsal column sensory axon regeneration after spinal cord injury would be ideal, but could be very challenging. Or, at least characterizing sensory axon regeneration into DREZ after dorsal root crush. If the results are negative, authors should discuss the limitation of manipulating this pathway alone in the CNS.

We understand that it would be interesting to test if fenofibrate can enhance regeneration past the DREZ or even promote some level of axon regeneration in the injured spinal cord. We considered these experiments, and opted to focus on axon regeneration along the dorsal root instead for the following reasons. First, it is extremely unlikely that fenofibrate treatment will enhance axon regeneration in the injured spinal cord. Indeed, the effect of fenofibrate on spinal contusion injury has been tested and proven ineffective to improve recovery (Almad et al. Exp Neurology 2011). Only a modest increase in histological tissue sparing was observed. Second, it has been shown that dorsal root axons regenerating along the root quickly stop at the scar-free DREZ, even after a nerve conditioning lesion (Chong et al., 1999; Zhang et al., 2007; Di Maio et al., 2011). This potent block on axon growth has been attributed to myelin-associated inhibitors and CSPGs. This suggests that fenofibrate, which would only target the satellite glial cells, is unlikely to be sufficient to promote axon regeneration through the dorsal root entry zone. Although various treatments have been found to promote growth past the DREZ and into the spinal cord, these are largely focused on treatments targeting the intrinsic growth capacity of sensory neurons, such as viral delivery of NGF, NT-3, ARTN or GDNF (Smith et al. Trends in Neurosci 2012). Recent studies have shown that combinatorial strategies that target myelin inhibitors and CSPG together with boosting the intrinsic axon growth capacity with GDNF enhanced regeneration of axon past the DREZ (Zhai et al. BioRxiv 2021). These studies highlight the importance of elevating intrinsic growth capacity to pass the DREZ. As we suggested in the Discussion section, fenofibrate could be considered in combinatorial therapies aimed at improving sensory function recovery after SCI. Future studies will be needed to test if targeting satellite glial cell with fenofibrate could synergize with neuron intrinsic approaches to enhance axon regeneration past the DREZ and also increase intraspinal axon regeneration. Such studies are, however, beyond the scope of the current manuscript.

3) Functional validation to address the role of PPARa signaling in satellite glial cells since the it was done with pharmacological manipulation.

We agree that this is an important point. We previously addressed the specificity of fenofibrate by demonstrating that PPARα in the DRG is highly enriched in satellite glial cells (Avraham et al. Nat Comm 2020). We showed that PPARα and PPARα target genes are upregulated in SGC but not in neurons after injury (Avraham et al. Nat Comm 2020). We also showed in an in vitro assay that fenofibrate does not promote growth in pure neuronal cultures, further supporting that PPARα is not expressed in neurons (Avraham et al. Nat Comm 2020). Immunostaining for PPARα demonstrated that PPARα is expressed in SGC but not neurons, and that neither injury nor fenofibrate treatment lead to PPARα expression in neurons (Avraham et al. Nat Comm 2020). Furthermore, a transcriptional profiling study of sensory neurons at single cell resolution (Renthal W. et al., Neuron, 2020) confirms that PPARα is not expressed in neurons, neither in naïve conditions nor following sciatic nerve crush injury. In the current study, we performed additional analyses to examine PPARα expression in other cells. First, we plotted all the cells expressing PPARα by cell type and injury condition. This analysis reveals that the majority of PPARα expressing cells are SGC in any injury condition (Figure 7C). Second, we present violin plots of PPARα and selected PPARα specific target genes, which demonstrates higher expression of PPARα in SGC compared to all other cells in the DRG (Figure 7 Supplement figure 1C). Given that fenofibrate is a selective activator of PPARα and does not target other PPAR isoforms (Lee CH, Olson P, Evans RM. Minireview: lipid metabolism, metabolic diseases, and peroxisome proliferator-activated receptors. Endocrinology. 2003 Jun;144(6):2201-7), we believe that the pharmacological manipulations presented here sufficiently address the role of PPARα signaling in satellite glial cells.

Reviewer #1 (Recommendations for the authors):In this manuscript, Avraham et al. report their results of profiling different cell types in DRG from mice with different types of injury. In general, injuries occurred in PNS (sciatic nerves or dorsal roots) trigger more drastic effects on nearly all DRG cell types, comparing to those applied to CNS (spinal cord), albeit with some exceptions. Among these responding cell populations is a subset of macrophage expressing a satellite glial cell (SGC) marker, and an another population of SGCs, although their lineage and role remain unknown. Furthermore, fatty acid biosynthesis and PPARa signaling pathways are up-regulated after sciatic nerve injury, but down-regulated after dorsal root injury (again these observations are not verified and the underlying mechanisms are elusive). Application of a PPARa agonist is able to elevate axon regeneration after dorsal root injury. In general, this manuscript has a large amount of data from bioinformatics analysis but with limited functional verifications. Thus their biological meaning is less clear.

Functional verifications were included in most figures. However, to address this point we have performed additional experiments and validations, as detailed below.

Figure 3: In addition to the immunofluorescence for macrophage and proliferation markers in DRG sections from all injury conditions, we now validated with qPCR the downregulation of selected cytokines and the upregulation of proliferation markers in macrophages following sciatic nerve crush injury (new Figure 3D).

Figure 4: In addition to the immunofluorescence of DRG sections showing co-expression of the SGC marker FABP7 and the macrophage marker CD68, we added a flow cytometry experiment of genetically labeled SGC (BlbpCreER:Sun1GFP), labeled with 3 different macrophage specific markers to validate the scRNAseq analysis (new Figure 4E). These new results support the notion that a subset of macrophages express glial properties at the protein level.

In Figure 3, the results imply different signaling involvement in DRG macrophages (cell cycle and DNA replication after SNC and steroid biosynthesis and glycolysis/gluconeogenesis pathways after DRC/SCI). Are these results due to differential resident/infiltrated macrophage in DRG after individual injury types?

We thank the reviewer for highlighting this point. Previous studies have indicated that the number of macrophages increases in the DRG after peripheral nerve injury but not dorsal root injury (Kwon et al. 2012). This increase in macrophages number after nerve injury results in part from proliferation of resident macrophages (Leonhard et al., 2002; Yu et al., 2020) and may also include local myeloid cell proliferation (Yu et al., 2020) and infiltration of a small number of blood-borne myeloid cells (Kalinski et al. 2020). We quantified all ki67 expressing cells in our scRNAseq, which demonstrates that the majority of cells proliferating following all injuries are macrophages, and that the number of proliferating macrophages is highest after sciatic nerve injury (Figure 3 K,L). Our results are consistent with these prior studies and suggest that the different signaling responses result largely from proliferation of macrophages after nerve injury.

It is also intriguing to note that all injuries down-regulate genes related to antigen processing and presentation (Figure 3). This seems an interesting observation as macrophages often exhibit pro-inflammatory responses to injury. These results should be verified with independent methods.

We agree and have performed a qPCR experiment to validate the downregulation of genes regulating antigen processing and presentation associated with class II major histocompatibility complex (MHC II) CD74, H2-Aa and Ctss (new Figure 3D). We also confirmed the downregulation of the cytokines Ccl2, Il1b and Tnf following sciatic nerve crush in qPCR experiments (new Figure 3D).

In Figure 4, the authors describe a subset of macrophages expressing glial markers whose numbers become increased after injury. Is it possible that these might be the macrophages with engulfed SGCs? To test this, perhaps the authors could compare the abundance of these type-specific RNAs or use other independent methods (with transgenic mice with GFP-labeled SGCs to see if any GFP signals are in these macrophages).

We agree that it is important to exclude the possibility that immune glial cells simply result from phagocytosis of satellite glial cells by macrophages. We have performed additional experiments and additional analyses that strongly support the existence of subset of macrophages with glial properties. We renamed these cells Imoonglia, to reflect their immune properties and their crescent shape morphology typical of SGC surrounding sensory neurons. First, we performed a flow cytometry experiment to show that a subset of genetically labeled SGC (BLBP-creER: Sun1 GFP) express the specific macrophage markers Cd11b, F4/80 and cd45 (new figure 4E). Second, we included additional analyses showing that these immune glia cells express progenitor cell markers (Dhh, *Sox2* and Foxd3), which are not expressed in macrophages (new figure 4C). Third, we performed a trajectory analysis demonstrating that Imoonglia express a transcriptome that position them between satellite glial cells and macrophages (new figure 4D). Fourth, we expanded the methods section to clarify that duplicate cells are filtered out from downstream analysis and also plotted total counts in all cell types (new figure 4—figure supplement 1B), further excluding the possibility that these cells are satellite glial cells engulfed by macrophages. We believe that these additional experiments and analyses strongly support the characterization of this Imoonglia cell population.

The results in Figure 5 suggest that SGCs represent different cell populations. Again, their biological meaning remains unknown. An obvious possibility is these clusters might reflect their different activation states. It might be useful to apply single cell trajectory analysis to assess their relationship.

We thank the reviewer for this suggestion and have performed trajectory analysis to assess the activation state and the relationship of the different SGC subtypes. The results indicate a trajectory starting from cluster 3, to cluster 2, then cluster 1 and finally cluster 4 (new Figure 5E). This is very interesting in light of our comparison of SGC clusters to astrocytes and Schwann cells, showing that cluster 3 most resembles astrocytes while cluster 4 mostly resembles Schwann cells (Figure 5H,I). The trajectory analysis comparing different cell lineage genes suggests that all SGC subtypes present the same activation state (Figure 5E). The biological function of these different SGC clusters awaits further in depth investigations that are beyond the scope of the current manuscript.

Figure 7, are the regeneration results after PPARa agonist comparable to those after sciatic nerve injury? Such information might provide insights as to its translational potential.

It has been shown that dorsal root axonal growth occurs at half the rate of peripheral axons (Oblinger and Lasek 1984; Wujek and Lasek, 1983). In the experiment presented in Figure 7E-G, we observed that fenofibrate treatment almost doubled the length of dorsal root axons, suggesting that activating SGC with fenofibrate can increase axon growth. In our previous study, we showed that deleting the enzyme Fasn, which is upstream of PPARα activation, specifically in SGC, decreases axon growth in the sciatic nerve by about half (Avraham et al. 2020). Altogether, these findings suggest that the lack of PPARα activation after dorsal root crush contributes to the low regeneration rates of axons in the dorsal root. We have edited the text in the Discussion section (p.22) to provide insights into the translational potential of fenofibrate.

Reviewer #2 (Recommendations for the authors):Suggested experiments:1) Impact on functional validation will be enhanced significantly if regeneration into and beyond DREZ can be assessed with fenofibrate, especially given the previously published results with SNC.

We understand that it would be interesting to test if fenofibrate can enhance regeneration past the DREZ or even promote some level of axon regeneration in the injured spinal cord. We considered these experiments, and opted to focus on axon regeneration along the dorsal root instead for the following reasons. First, it is extremely unlikely that fenofibrate treatment will enhance axon regeneration in the injured spinal cord. Indeed, the effect of fenofibrate on spinal contusion injury has been tested and proven ineffective to improve recovery (Almad et al. Exp Neurology 2011). Only a modest increase in histological tissue sparing was observed. Second, it has been shown that dorsal root axons regenerating along the root quickly stop at the scar-free DREZ, even after a nerve conditioning lesion (Chong et al., 1999; Zhang et al., 2007; Di Maio et al., 2011). This potent block on axon growth has been attributed to myelin-associated inhibitors and CSPGs. This suggests that fenofibrate, which would only target the satellite glial cells, is unlikely to be sufficient to promote axon regeneration through the dorsal root entry zone. Although various treatments have been found to promote growth past the DREZ and into the spinal cord, these are largely focused on treatments targeting the intrinsic growth capacity of sensory neurons, such as viral delivery of NGF, NT-3, ARTN or GDNF (Smith et al. Trends in Neurosci 2012). Recent studies have shown that combinatorial strategies that target myelin inhibitors and CSPG together with boosting the intrinsic axon growth capacity with GDNF enhanced regeneration of axon past the (DREZ Zhai et al. BioRxiv 2021). These studies highlight the importance of elevating intrinsic growth capacity to pass the DREZ. As we suggested in the Discussion section, fenofibrate could be considered in combinatorial therapies aimed at improving sensory function recovery after SCI. Future studies will be needed to test if targeting satellite glial cell with fenofibrate could synergize with neuron intrinsic approaches to enhance axon regeneration past the DREZ and also increase intraspinal axon regeneration. Such studies however are however, beyond the scope of the current manuscript.

2) Figure 3e and 3g have different markers. Validation will be enhanced if selected markers in Figure 3e is confirmed with immunostaining.

We now matched the gene and protein name of Ki67 to clarify. CD68 in Figure 3G is used to identify macrophages as over +90% of macrophage express CD68 (Figure 3A). The heat map in Figure 3E is focused on macrophage proliferation markers, cell division markers, cytokines and chemokines. This has been highlighted appropriately now. As there are no good antibodies for cytokines for immunostaining, we chose cytokines that are specifically expressed in macrophages (Ccl2, Il1B TNF) and validated their expression following sciatic nerve crush in qPCR experiments (new Figure 3D).

Our analysis suggests that SNC and DRC lead to greater gene expression changes compared to SCI (Figure 3B), but, interestingly, the pathway analysis shows high similarity between DRC and SCI (Figure 3D).

Suggested discussions, clarification or corrections:1) Using 1 biological replicate for SNC vs. 2 for other conditions seems unbalanced. Ideally use 2 replicates per condition, and at least discuss implications on using 1 replicate for SNC. For instance, what is the variability among the replicates for data presented in Figure 1f,g?

We agree with the reviewer’s comment for having 2 replicates for each condition. This is what we aimed for. However, the reason that in this study we included 1 replicate for SNC is the following: we initially combined the 2 replicates for SNC from our previous study (Avraham, et al. Nat Comm 2020) with one new SNC replicate. Combining these two different datasets complicated the analysis, as there was significant batch effect, that likely resulted from the different chemistry of the library kits used (V2 vs. V3). The number of cells captured and the depth of analysis (mean reads per cell) were also different. However, analysis of the enriched biological pathways in the 2 datasets revealed high similarity and was thus suggesting that the biological meaning remains. The variability among replicates for the data presented in Figure 1F,G is shown in Figure 1, Figure supplement 1A.

2) Authors indicated that the magnitude of expression changes using number of differentially expressed genes. With Supplemental Table 2, the cut off was based on 0.05 p value instead of FDR corrected (FDR step up). With FDR corrected p-value, the numbers become 614 (SRC vs. NAI), 222(SCI vs. NAI) and 1276 (SNC vs. NAI). Please change it accordingly as the multiple comparison is necessary for sequencing data.

We thank the reviewer for highlighting this point. We corrected the p-value to FDR and adjusted all DEG analysis including heatmaps and source tables.

3) Is PPARa pathway enriched in cluster 6 SGCs, which increases after SNC?

We thank the reviewer for this important question. The PPARα pathway is not specifically enriched in SGC subcluster 6 (new Figure 6J). The expression of the PPARα transcript was observed in all SGC clusters, including cluster 6, with highest distribution in subclusters 2 and 3 (new Figure 7D).

4) The discussion on immune glia cells can be more explicit, even though it is understood that this new cell type may not squarely fit with either macrophages or SGCs. Please discuss if this been implicated in the published literature.

We agree and have revised the text to better explain what these cells may represent. There is indeed evidence in the literature for the existence of cells with morphology similar to SGC, with close apposition to the neuron and with expression of immune markers. Van Velzen et al. (2009) described the presence of immune markers in human SGC. This was based only on cell localization without co-staining with SGC markers. Jasmin et al. (2010) also found that some SGC express CD45, but unlike in humans, the CD45 positive cells were only present after nerve injury. Donegan et al. 2013 observed that the expression of immune markers (CD45 and ED2) in SGC occurred 2 and 4 days post nerve injury. While some were co-localized with proliferating nuclei, others were not. Huang et al. 2021 also found that a subset of SGC showed CD45 immunoreactivity. A recent comparative analysis of sympathetic and sensory SGC suggests a shared populations of satellite glia enriched in immune-response genes (Mapps et al. BioRxiv 2021). In another single cell based analysis of SGC , a cluster was excluded from further analyses due to relatively high expression of reactive markers, including interferon-related genes (van Weperen, Glia 2021). Our analysis, combines scRNAseq, flow cytometry and high-resolution co-immunostaining to provides a more comprehensive characterization of this rare cell type, which we named Imoonglia, to reflect their immune properties and their crescent shape morphology typical of SGC surrounding sensory neurons. What could be the origin of these cells? Our analysis suggests that Imoonglia do not proliferate, but express progenitor markers. Whether immune cells move to a peri-neuronal position and become Imoonglia or whether SGCs transition from a progenitor state to acquire an immune signature in stressful condition such as after nerve injury remains to be tested. It is tempting to speculate that Imoonglia may function in immune surveillance in the DRG. Indeed, SGC play a role in viral infection and may restrict the local diffusion of viruses such as herpes simplex virus, varicella zoster virus, HIV-1 and Hemagglutinating encephalomyelitis virus belongs to the family of coronavirus (Hanani review 2020).

5) Please clarify if the data on SGCs overlap with those from Avraham et al., 2020 (i.g. same data set)?

In this study we generated new sample after SNC, which is distinct from the 2 replicates for SNC from in our previous study (Avraham, et al. Nat Communications 2020). However, we confirmed that in the current and previous dataset (Avraham et al.,2020), the enriched biological pathways between the 2 datasets were very similar and thus ensuring that the biological meaning remains (Author response image 1) .

**Author response image 1. sa2fig1:** Enriched pathway (KEGG 2019) in SGC 3 days after SNC in our previous data set (2020) and new data set (2021).

6) "Here we showed that this PPARa signaling pathway is not activated in SGC after central injures." Is DRC a peripheral or central injury?

We thank the reviewer for highlighting this point and we have revised the text in Results section to better explain the various injuries (p. 6). Unlike SCI injury, dorsal root injury damages centrally projecting sensory axons in the PNS, without causing an impassable glial scar. But axons grow at ½ the speed in the dorsal root until they stop at the scar-free dorsal root entry zone (DREZ). This model provides a tool to see if we can increase axon growth capacity without the complication of the glial scar or the DREZ.

The sentence on p. 22 has been rephrased to “….to centrally projecting axons (SCI or DRC)”

7) For DRC experiment, was DRC only performed for L4, but both L4 and L5 DRG were collected for scRNA seq?

For DRC, both L4 and L5 roots were crushed and both L4 and L5 DRG were collected. The distance from the crush site is 4-5mm to L4 and 7-8mm to L5. We have revised the method section on p. 28 to clarify this point.

8) It seems that the least differentially expressed genes are associated with the most abundant cell types sequenced (SGCs and macrophages). Is this due to a higher sample rates, thus more reproducible/reliable data set? And vice versa?

We thank the reviewer for this observation and agree for the possibility that the number of DEG is determined by the sample size. A more reasonable option is that when a cell population has several subpopulations, like we see for SGC, if the subpopulations respond differentially, then the number of DEG will decrease when DEG are calculated for the total SGC (i.e., same gene increase in one subpopulation but decrease in another). To test this possibility, we calculated the DEG for every SGC subpopulation and indeed found that every subtype responds differently (figure 6- Source Data 1), with cluster 4 showing the highest transcriptional changes (now added as new Figure 6E).

9) "Whether the different regenerative capacities after peripheral or central injury result in part from a lack of response of macrophages, satellite glial cells (SGC) or other non-neuronal cells in the DRG microenvironment remains largely unknown." Can also be an altered response.

We agree and have edited this statement, which now reads as ”Whether the different regenerative capacities after peripheral or central injury result in part from a lack or an altered response of macrophages, satellite glial cells (SGC) or other non-neuronal cells in the DRG microenvironment remains largely unknown”.

10) Even though fenofibrate may have minimal activity towards PPARa, a previous paper on the neuronal role of the γ isoform is relevant and should be discussed: Lezana et al., Dev Neurobiol 2016. Axonal PPARa promotes neuronal regeneration after injury.

We agree and have now cited this paper in the discussion, p.22.

11) "Sciatic nerve crush injures approximately half the axons projecting into the peripheral nerves…" Please clarify why this is the case. The percent of DRG neurons lesioned under any injury paradigms (e.g. SNC vs DRC) may impact the injury response of the microenvironment. Also, SCI only injures large diameter DRG neurons, which may explain some of the differences. Would injury level (T9/10) or the distance from DRG neuronal cell bodies also play a role (e.g. DRC is much closer to the cell bodies)?

The sciatic nerve is composed of axons projecting from sensory neuron soma residing in multiple lumbar DRG and SNC results in ~50% of lumbar DRG neurons being axotomized (Laedermann et al., 2014; Rigaud et al., 200, Renthal 2020). SNC injury is also more distant from the cell body compared to DRC (20mm in SNC to L4 and L5 in SNC vs 4-8mm 20mm in DRC). These differences may impact the injury responses of the microenvironment.

We also agree that the percent of DRG neurons lesioned under any injury paradigm and the distance of the injury from the DRG may impact the injury response of the microenvironment. We have included this point in the results (p. 6) and discussion (p. 19). However, all three injury paradigms are widely used models to study the mechanisms promoting axon regeneration.

12) "Dorsal column lesion of the spinal cord damages the ascending axon branches of most large diameter neurons and leaves the descending axon branches in the spinal cord intact." Needs qualifier, as this statement requires certain conditions to be met (e.g., in this case, T9/10 SCI, L4/5 DRG).

We agree and have provided the additional information in the method section on p. 28. The dorsal column was injured at the T9/10 level and L4/5 DRGs were collected for the experiments.

13) "…1 biological replicate for SNC (Figure S1d), with an average of 45,000…" Is Figure S1d the right panel here?

Thank you for the correction. Corrected to figure 1—figure supplement 1A

14) Figure 3 yellow or orange?

Orange, corrected in figure legend

15) In text, reference to phagosomes should be Figure 3d, not 4d.

Thank you for the correction.

16) "video 1,2". There is video 2?

Thanks for the correction, only one video is provided.

17) Please indicate p value when discussing differentially expressed genes, when appropriate.

Description of FDR and Fold change cutoffs of DEG was added to the text and figures when appropriate.

18) PT53 should be TP53.

Thank you, corrected.